


# Representation of solar tides in the stratosphere and lower mesosphere in state-of-the-art reanalyses and in satellite observations

Takatoshi Sakazaki[1, 2, 3], Masatomo Fujiwara[4], and Masato Shiotani[3]

[1]International Pacific Research Center, University of Hawai'i at Manoa, Honolulu, HI 96822, USA
[2]Japan Society for Promotion of Science Overseas Research Fellow, Tokyo, 102-0083, Japan
[3]Research Institute for Sustainable Humanosphere, Kyoto University, Uji, 611-0011, Japan
[4]Faculty of Environmental Earth Science, Hokkaido University, Sapporo, 060-0810, Japan

*Correspondence to*: Takatoshi Sakazaki (tsakazak@hawaii.edu)

**Abstract.** Atmospheric solar tides in the stratosphere and the lower mesosphere are investigated using temperature data from five state-of-the-art reanalysis data sets (MERRA-2, MERRA, JRA-55, ERA-Interim and CFSR) as well as TIMED/SABER and Aura/MLS satellite measurements. The main focus is on the period 2006-2012 during which the satellite observations are available for direct comparison with the reanalyses. Diurnal migrating tides, semidiurnal migrating tides, and nonmigrating tides are diagnosed. Overall the reanalyses agree reasonably well with each other and with the satellite observations for both migrating and nonmigrating components including their vertical structure and the seasonality. However, the agreement among reanalyses is more pronounced in the lower stratosphere and relatively weaker in the upper stratosphere and mesosphere. A systematic difference between SABER and reanalyses is found for diurnal migrating tides in the upper stratosphere and the lower mesosphere, specifically that the amplitude of trapped modes in reanalyses is significantly smaller than that in SABER, although such difference is less clear between MLS and reanalyses. The interannual variability and the possibility of long-term changes of migrating tides are also examined using the reanalyses during 1980-2012. All the reanalyses agree in exhibiting a clear quasi biennial oscillation (QBO) in the tides, but the most significant indications of long-term changes in the tides represented in the reanalyses are most plausibly explained by the evolution of the satellite observing systems during this period. The tides are also compared in the full reanalyses produced by the Japan Meteorological Agency (i.e. JRA-55) and in two parallel data sets from this agency: one (JRA-55C) which repeats the reanalysis procedure but without any satellite data assimilated, and one (JRA-55AMIP) that is a free-running integration of the model constrained only by observed sea surface temperatures. Many aspects of the tides are closer in JRA-55C and JRA-55AMIP than these are to the full reanalysis JRA-55, demonstrating the importance of the assimilation of satellite data in representing the diurnal variability of the middle atmosphere. In contrast to the assimilated data sets, the free running model has no QBO in equatorial stratospheric mean circulation and our results show that it displays no quasibiennial variability in the tides.





## 1 Introduction

Atmospheric solar tides are global-scale inertia-gravity waves with periods that are integer fractions of a solar day (Chapman and Lindzen, 1970). They are primarily driven by diurnally varying diabatic heating, such as the absorption of solar radiation by tropospheric water and stratospheric ozone, and the latent heat release associated with tropical convection. The diurnal
($S_1$) and semidiurnal ($S_2$) variations around the globe can be decomposed into zonal harmonics with the "migrating" (Sun-synchronous) components for the $S_1$ / $S_2$ tides represented by westward propagating wave number one / two. The remainder of the tidal zonal harmonics are "non-migrating components" and are excited mainly by zonally-asymmetric variations in (local time) heat sources or topography. Tides propagate vertically with amplitudes typically reaching a maximum in the mesosphere and lower thermosphere (MLT) region. There have been many studies of the tides in MLT as seen in ground-
based measurements, satellite measurements, and numerical simulations (e.g., Lieberman, 1991; Hagan et al., 1995; Forbes and Wu, 2006; Zhang et al., 2006; Ward et al., 2010).

The amplitudes of tidal variations in the region from the troposphere to the lower mesosphere are generally smaller than in the MLT and so fewer studies have investigated the tides in this lower altitude region. Nevertheless, tidal variations in the
lower atmosphere are worth investigating not only because they provide a 'lower boundary condition' for tides in upper air but also because the tide and the resultant diurnal cycle in stratospheric ozone (Sakazaki et al., 2013a) need to be considered when constructing a homogenized data set of temperature and ozone from different satellites with different measurement local times (e.g., Zou et al., 2014; WMO, 2014; Nash and Saunders, 2015; Sakazaki et al., 2015c). Also it is now established that tides excited in the stratosphere can play a significant role in tropospheric meteorology, particularly in the diurnal cycle
of tropical rainfall (Woolnough et al., 2004; Sakazaki et al., 2017).

The global pattern of stratospheric tides was investigated based on temperature data from ground-based (radiosondes, lidars) and satellite measurements (e.g., Wallace and Hartranft, 1969; Keckhut et al., 1996; Leblanc et al., 1999; Xu et al., 2009; Mukharov et al., 2009; Huang et al., 2010; see also  Sakazaki et al. 2012 and references therein). However, the available
direct measurements have important limitations in temporal and spatial coverage. For example, SABER, which is so far the most commonly-used data set for tidal studies, has difficulty detecting tides in regions poleward of ~50°. Meteorological reanalyses that provide temporally and spatially homogeneous data over the globe could be a useful data set for a tidal study. Using a fixed assimilation–forecast model system, the reanalyses provide best estimates of past atmospheric states in many dynamical variables. Currently available reanalyses from major centres provide estimates of atmospheric variables from the
surface to the upper stratosphere or the lower mesosphere, with time resolution of 3 or 6 hours.

Reanalysis for relatively high-frequency features such as the tide are particularly challenging in the region above the usual ~10 hPa upper boundary of conventional balloon soundings.  Above the middle stratosphere the reanalyses must rely on



direct observations only from satellites (e.g., Fujiwara et al., 2017) which have limited local-time/space coverage and so the reanalysis representation of tides in this region may be particularly dependent on the tidal simulation in the forecast models employed.

Previous studies evaluating the representation of solar tides in the stratosphere and the lower mesosphere in reanalyses have mainly considered only $S_1$ (24 hr) tides. An early study by Swinbank et al. (1999) investigated the $S_1$ migrating tide in the stratosphere as represented in the Goddard Earth Observing System (GEOS), Version 2 analysis data (one of the predecessors of MERRA and MERRA2 reanalyses). They found that the GEOS-2 tidal amplitude in the free-running model is reduced by assimilating satellite data, particularly data from the stratospheric sounding unit (SSU). Sakazaki et al. (2012,

hereafter referred to as S12) compared $S_1$ migrating tides in the stratosphere by using data from TIMED/SABER and six reanalysis data: MERRA, ERA-Interim, CFSR, JRA25/JCDAS, NCEP1 and NCEP2. They found that the overall latitude-altitude structure and its seasonality was reproduced qualitatively by the newer three reanalyses (MERRA, ERA-Interim and CFSR), but the amplitude in reanalyses was 30-50% underestimated in the upper stratosphere and lower mesosphere.

Only a few studies have examined $S_2$ tides in reanalyses (although the solar $S_2$ surface pressure oscillation has been more extensively studied, e.g., Ray and Ponte, 2003; Saha et al., 2010; Díaz-Argandoña et al., 2016; Hamilton and Sakazaki, 2017). Hsu and Hoskins (1989) and Kohyama and Wallace (2014) derived $S_2$ migrating tides in the stratosphere by using the ECMWF operational analysis and the ERA-Interim, respectively. Li et al. (2015) used CFSR reanalysis to examine the seasonality of $S_2$ migrating tides. Kopp et al. (2015) compared the $S_2$ tides derived from lidar measurements over

Kühlungsborn (54°N, 12°E) with those from MERRA reanalysis. Note that no inter-comparison of the $S_2$ tides as represented in different reanalyses has so far been performed. Note also that nonmigrating tides have not been examined with reanalysis data, as far as the authors are aware.

Since the study of S12, several new reanalysis data sets have been released including MERRA-2 and JRA-55. The present

study is a follow-up of S12 including these new reanalyses and extending the analysis to the $S_2$ migrating tides and to nonmigrating tides. Also the Japan Meteorological Agency has produced a unique resource in which the full state-of-the-art JRA-55 reanalysis is supplemented with two additional global data sets  (collectively called `JRA-55 family'): JRA-55C, which assimilates only conventional surface and balloon sounding observations, and JRA-55AMIP, which employs a free-running version of the forecast model. The comparison of JRA-55 family members enables us to investigate the effects of

data assimilation on the representation of tides in the global data sets. Data from Aura/MLS (only assimilated in MERRA-2) will be also analyzed as a measure of $S_1$ migrating tide.

The remainder of the manuscript is organized as follows. Section 2 describes the reanalysis and observational data sets employed, while Section 3 describes our method to extract tidal components. Section 4 shows the results for $S_1$ migrating



tides, $S_2$ migrating tides, and nonmigrating tides (mainly for $S_1$). Section 5 examines the long-term changes in migrating tides as represented in the reanalyses over the last three decades, while major findings are summarized in Section 6. This work described in the present paper contributes to the SPARC Reanalysis Intercomparison Project (S-RIP), Chapter 11: "Upper stratosphere and lower mesosphere" (see Fujiwara et al., 2017 for details about the S-RIP).

**2 Data sets**

We analyze and compare data from global reanalyses and two satellite observational data sets: SABER (not assimilated in any reanalyses) and Aura/MLS (only assimilated in MERRA-2). In Section 5 below we will intercompare several reanalyses over a long period (1980-2012) that includes most of the modern era in which satellite radiances have been assimilated. However, our detailed evaluation will focus on the 7-year period 2006-2012 during which it seems the satellite data sources

used in the global assimilations were fairly stable (e.g. Kawatani et al., 2016 and Fujiwara 2017) and during which we have two other satellite data sets (SABER and Aura/MLS) not included in most of the assimilations and thus providing independent estimates of the diurnal variability of temperature in the stratosphere and mesosphere. Note that all the reanalysis data sets employed extended over the full 1980-2012 period with the exception of CFSR whose integration with the original CDAS-T382 system ends in December 2010 (see Fujiwara et al., 2017).

**2.1 Reanalyses**

We compare results in satellite data sets with those from seven different global gridded data sets produced at major meteorological centres. Five of these are standard state-of-the-art global atmospheric reanalyses: (1) MERRA-2 (Gelaro et al., 2017), (2) MERRA (Rienecker et al., 2011), (3) JRA-55 (Kobayashi et al., 2015), (4) ERA-Interim (Dee et al., 2011), and (5) CFSR (Saha et al., 2010). We will not consider here the JRA-25, ERA40 and NCEP1/2 reanalyses, which are the

predecessors of JRA55, ERA-Interim and CFSR, respectively. S12 showed that the global structure and seasonality of the $S_1$ migrating tide represented in JRA25 or NCEP1/2 were less consistent with available observations than were the newer reanalyses data sets.

In addition to the five full reanalyses we also analyze the tides in two other gridded products produced by the Japan

Meteorological Agency that are parallel to their full JRA-55 reanalysis. One (JRA-55C) repeats the reanalysis procedure but assimilating surface and upper-air conventional data but not satellite data, and the other (JRA-55AMIP) is a free-running integration of the forecast model constrained only by observed sea surface temperatures (Kobayashi et al., 2014). Here we refer to the three JMA data sets (JRA-55, JRA-55C and JRA-55AMIP) as the "JRA-55 family". The comparison of these family members will help us examine the effects of data assimilation on the representation of the solar tides.



Each reanalysis system is comprehensively described by Fujiwara et al. (2017) and thus only key aspects are summarized here (see also **Table 1**). Data are available 3-hourly at 0000, 0300, 0600, 0900, 1200, 1500, 1800 and 2100 UTC for MERRA and MERRA-2, and 6-hourly at 0000, 0600, 1200 and 1800 UTC for the remaining data sets. Data provided on the output pressure levels are used for MERRA and MERRA-2, CFSR and JRA55-AMIP, with the number of levels being 42

(up to 0.1 hPa), 37 (up to 1 hPa) and 37 (up to 1 hPa), respectively. For JRA55, JRA55C and ERA-Interim, we interpolated data provided on model levels onto the 42 MERRA and MERRA-2 pressure levels up to 0.1 hPa.

## 2.2 Satellite measurements

### 2.2.1 SABER

The SABER instrument is onboard the TIMED satellite, which was launched on December 7, 2001 (Russell et al., 1999). It

measures $CO_2$ infrared limb radiance to retrieve the kinematic temperature profiles between 20 km and 120 km (Remsberg et al., 2008). Data are continuously obtained between 53°S and 53°N. The TIMED satellite is not in a Sun-synchronous orbit, and the local time of SABER measurements changes by about 12 min per day, meaning that a full diurnal cycle (24 hours in local time) is covered over a period of 60 days using ascending and descending nodes. Note that data are not acquired by SABER near local noon. The vertical resolution of the measurements is ~2 km.

In our study, version 2.0 temperature data on pressure levels are analyzed for January 2006 through December 2012 (S12 analyzed version 1.07 data). As described by Sakazaki et al. (2015b), before the further analysis, data were averaged in bins of 15° in longitude, 5° in latitude, and 2 km in log-pressure vertical coordinates, for each day and for each ascending and descending node. We emphasize that SABER data were not assimilated into any of the reanalyses used in this study.

### 2.2.2 Aura/MLS

The MLS instrument is onboard the Aura satellite, which was launched July 2004. It uses the microwave limb sounding technique to observe atmospheric dynamical parameters and chemical constituents (Waters et al., 2006). The Aura orbit is sun-synchronous at 705 km altitude with 98° inclination, with the equator-crossing local time being 1:45 pm for the

ascending nodes. The MLS fields of view look in the forward direction (almost north-south direction) and vertically scan the limb of the atmosphere. In the tropics (10°S-10°N), the actual measurement local times were 13:45 and 01:45 on average for the ascending and descending nodes, respectively.

MLS temperature is retrieved from bands near $O_2$ spectral lines, for the region between 261 and 0.001 hPa pressure levels

with the vertical resolution of 3-6 km. In this study, version 3.3 data are utilized after a data-screening performed based on



the criteria shown by Livesey et al. (2011). The MLS measurements were assimilated into the MERRA-2 data set at pressure levels less than 5 hPa (after 2004, Gelaro et al., 2017), but not into any of the other reanalyses that we employed.

## 3 Analysis Methods

5 ### 3.1 Migrating and nonmigrating tides

In this study, the (1) diurnal ($S_1$) migrating tide, (2) semidiurnal ($S_2$) migrating tide, and (3) nonmigrating tides are extracted and diagnosed individually. The analysis procedure to extract these three components basically follows the method proposed by Sakazaki et al. (2015b) as briefly explained below.

10 First, all diurnal variations, which includes both migrating and nonmigrating components, are calculated based on universal time (UT) as follows. For SABER data, since 24 local times are covered by 60-day measurements by ascending and descending nodes, a time-series of 60-day running mean that can be regarded as the daily-mean is calculated for each latitude-longitude bin; this is then subtracted from the original temperatures for each day, for each bin, and for each descending and ascending node to produce the anomaly from the daily-mean. These anomaly temperatures are binned and 15 averaged into hourly UT time bins to obtain 1-hourly diurnal variations. For reanalyses, 3- or 6-hourly diurnal variations in UT are extracted at each grid point by the composite analysis based on UT after the subtraction of the daily-mean (in fact we downloaded and analyzed 'diurnal monthly mean (monthly mean for each UTC snapshot)' data provided by each reanalysis centre (e.g., for JRA-55, diurnal monthly mean data are the monthly averages for 0000, 0600, 1200, and 1800 UTC, respectively)). Obviously the 6-hourly data (ERA-Interim, JRA-55 and CFSR) cannot resolve $S_2$ at each grid point; but the 20 'migrating component' of $S_2$ can be extracted by using data at grid points on the same latitude belt, as explained in the following (cf., Ray and Ponte, 2003; Díaz-Argandoña et al., 2016; Hamilton and Sakazaki, 2017).

Next, by averaging data at the same local time (LT) for each latitude band, migrating tides that are a function of LT are calculated; for example, for 6-hourly reanalyses, data at 0000 LT is the average of data points at (0000 UT, 0°E) (0600 UT, 25 90°E) (1200 UT, 180°E) (1800 UT, 270°E). Then, the harmonic fitting is performed for the diurnal variations in LT to extract the migrating $S_1$ and $S_2$ components. Finally, nonmigrating tides are calculated by subtracting migrating tides (for reanalyses, the $S_1$ plus $S_2$ migrating tides are used for actual calculation) from the whole tidal variations.

For nonmigrating tides, the zonal wavenumber decomposition is also applied for the $S_1$ component, following the method 30 proposed by Dai and Wang (1999). Before the analysis, the tidal component ($X$) at any longitude ($\lambda$), latitude ($\theta$) and vertical pressure level ($z$) was decomposed into the symmetric ($X_S$) and anti-symmetric ($X_A$) components with respect to the equator as,



$$X(\lambda,\theta,z) = X_S(\lambda,\theta,z) + X_A(\lambda,\theta,z) \quad (1)$$

, where

$$X_S(\lambda,\theta,z) \equiv \frac{1}{2}\{X(\lambda,\theta,z) + X(\lambda,-\theta,z)\} \quad (2)$$

$$X_A(\lambda,\theta,z) \equiv \frac{1}{2}\{X(\lambda,\theta,z) - X(\lambda,-\theta,z)\} \quad (3)$$

### 3.2 Difference between ascending minus descending nodes for MLS (A-D difference)

The difference between ascending (A) and descending (D) nodes of MLS temperature measurements is calculated in the tropics (10°S-10°N) as a measure of tidal amplitude (hereafter referred to as A-D difference). Since the measurement times are fixed and are 12 hr apart in local time (i.e., ~1:45 am/pm), the zonal mean A-D difference can be caused by odd harmonics of the migrating tides (24-h, 8-h components etc). In addition, because the $S_1$ component is predominant over the higher-order harmonics (e.g., the ratio of 8h- to 24h- components in SABER data was <30% for most parts in the tropical stratosphere and in the lower mesosphere), the A-D difference can be mostly attributed to the $S_1$ migrating tide. The same quantity is calculated for the other data sets as well (i.e., SABER and reanalyses) by using the migrating tides (i.e., diurnal variations in LT) in each data set (for reanalyses, migrating tides reconstructed from $S_1$, $S_2$ and terdiurnal harmonic components are considered).

### 4 Results

#### 4.1 Diurnal ($S_1$) migrating tide

**Figures 1 and 2** show the latitude-altitude distribution of amplitude and phase, respectively, for annual-mean $S_1$ migrating temperature tides computed from SABER data and from the various reanalyses during 2006-2012. **Figure 3** compares the vertical profile of amplitude and phase averaged over 15°S-15°N (tropics) and 30°N-45°N (mid-latitudes). All data sets show that the tidal amplitude increases with altitude in the tropics (up to ~4K in the lower mesosphere in the SABER data, somewhat less in the various reanalyses). The amplitude has maxima in the upper stratosphere (45-50 km or 1 hPa) ~3.5 K for SABER (again somewhat less in the reanalyses) in the extratropics of both hemispheres. Over the tropics, the phase shows a downward progression (except for SABER at 40-55 km; see below for further discussion). At extratropical latitudes, on the other hand, the phase is almost constant around 18 LT except for the region above ~55 km (0.3 hPa).





There is quite a good agreement among the full reanalysis data sets below ~45 km, while the spread among the reanalysis results becomes larger above the upper stratosphere. It is inferred that the reanalyses may be well constrained by satellite measurements up to the upper stratosphere, while being somewhat more model-dependent in the lower mesosphere.

5    Apart from the difference among different reanalyses, we see a systematic difference between SABER and all the reanalyses both for amplitude and phase above 40 km. Notably: (1) the amplitude in SABER is ~1 K larger than that in the reanalyses, (e.g., see the extratropical maximum at ~45 km in **Fig. 3c**) and (2) the phase is locally constant (or shows a upward progression) in the tropics for SABER at 40-55 km while it shows a continuous downward progression for most reanalyses. The present SABER results including the extratropical maxima at 45-50 km and the phase stagnation at 40-50 km are 10    quantitatively consistent with previous studies using SABER, even though earlier investigators used a different procedure to extract tides (Mukhtarov et al., 2009; Xu et al., 2009; Huang et al., 2010).

In **Fig. 3** it is apparent that the JRA-55C and JRA-55AMIP results for the $S_1$ migrating tide stand out from those obtained with the full reanalyses. Notably the JRA-55C and JRA-55AMIP results are close together and differ substantially from the 15    JRA-55 results. The amplitude in JRA-55C and JRA-55AMIP is no larger than that in JRA-55 for the entire stratosphere and is substantially smaller in some regions. This suggests that, contrary to the finding by Swinbank et al. (1999), the assimilation of satellite measurements does not act to damp the tidal amplitude in JRA-55 at least for the recent period (Swinbank et al. analyzed data in early 1990s).

20    We analyzed these results further with some guidance from the expectations of so called "classical tidal theory" which solves for the linear response of the global atmosphere to monochromatic heating ignoring mean winds and horizontal temperature gradients in the mean state. The classical tidal theory equations are separable in the zonal, vertical and meridional directions and conventionally the solutions are written as the product of zonal harmonics and meridional modes known as Hough functions (e.g. Chapman and Lindzen, 1970) As shown by Sakazaki et al. (2013b), the $S_1$ migrating tide in 25    the stratosphere can be reasonably well represented by a superposition of only a few (~4) Hough modes each of which has its own vertical propagation characteristics. For the annual-mean tidal temperatures which are almost symmetric about the equator (**Fig. 1**), even the two gravest symmetric Hough modes ((1, 1) mode and (1, -1) mode shown in **Fig A1**) are enough to represent the overall structures. That is, the $S_1$ migrating tidal temperatures ($T_{S1\text{-}mig}$) determined from SABER and each of the reanalyses are approximated as,

30    $$T_{S_1-mig}(\theta,z,t) = \overline{T}(\theta,z)\cos(\omega(t-\overline{\alpha}(\theta,z))) \cong \sum_{n=1}^{2} \widetilde{T}_n^1(z)\Theta_n^1(\theta)\cos(\omega(t-\widetilde{\alpha}_n^1(z))), (4)$$

where $t$ is local time (hr); $\overline{T}$ and $\overline{\alpha}$ are amplitude (K) and phase (LT), respectively at each latitude-pressure level; $\omega = 2\pi/24$ (hr$^{-1}$); $\widetilde{T}_n^1$ and $\overline{\alpha}_n^1$ are the amplitude (K) and phase (LT) of the $n$-th Hough mode ($\Theta_n^1$; in this case, $n$ = 1 is the (1, 1) mode



and $n = 2$ is the (1, -1) mode). Note that the equatorially-trapped (1,1) mode is associated with vertical phase propagation while the (1,-1) mode represents disturbances we expect to be vertically-trapped.

**Figure 4** shows the vertical profile of amplitude and phase of the two modes (i.e., $\tilde{T}_n^1(z)$ and $\bar{\alpha}_n^1(z)$ of Equation (4), respectively). For the propagating (1,1) mode, the amplitude grows exponentially with increasing altitude and the phase shows a downward progression. The vertical wavelength is ~25 km, quite consistent with the prediction by classical tidal theory (~28 km; Chapman and Lindzen, 1970). For the trapped (1,-1) mode, the amplitude is localized around the peak ozone heating region (~50 km) and the phase is almost constant with altitude around at 18 LT. Notably the systematic difference between SABER and reanalyses seen in **Fig. 3** is projected mostly onto the amplitude of trapped mode (**Fig. 4c**); the amplitude of trapped mode in reanalyses is 1.5-2.5 K, significantly smaller than that in SABER (3-4 K). For the propagating mode, by contrast, there is no clear systematic difference between SABER and reanalyses (**Figs. 4a and 4b**). Because the magnitude of trapped mode is smaller in reanalyses compared to SABER, the amplitude is small at all latitudes and the phase can propagate vertically in the tropics (i.e., in SABER, the phase is almost constant at 40-55 km affected by the strong trapped mode; **Fig. 3**). The magnitude of trapped mode in SABER is consistent with the analysis by Mukhtarov (2009) (~4 K peak both in March and July). These findings imply two possible reasons for the SABER-renalyses difference: (1) (if SABER is `true') the ozone heating, which is largely responsible for the trapped mode in the upper stratosphere, may be underestimated in reanalyses, or (2) (if reanalyses are `true') the SABER might have any bias that is dependent on local time and has a similar latitudinal structure similar to the trapped mode (i.e., almost constant with latitude).

To supplement the above discussion concerning the $S_1$ migrating tide, we examined the A-D difference (i.e., 13:45 LT minus 01:45 LT) in MLS temperature measurements. As mentioned in Section 2, the zonal-mean A-D difference is expected to result from $S_1$ migrating tides (if there is no 'instrumental' bias between A and D profiles). The vertical profile of the 1345 LT minus 0145 LT difference averaged over 10°S-10°N is shown in **Fig. 5** for the MLS determinations, and for the SABER data and each of the reanalyses. The A-D difference in MLS and all the reanalyses, but not SABER, basically changes its sign vertically, with its absolute value increasing with altitude. This feature means that the amplitude increases with altitude and the phase shows a vertical progression. The profile by SABER, by contrast, is mostly positive over the entire upper stratospheric region; this corresponds to the fact that the phase from SABER shows little vertical progression at 40-55 km (**Fig. 3b**).

It may be worth comparing the present results with previous findings, especially for the upper stratosphere and the lower mesosphere. Wu et al. (1998) analyzed temperature measurements from the MLS onboard the UARS. In the tropics (15°S-





15°N), they showed that the amplitude of $S_1$ migrating tides is ~1 K at 1 hPa (see their Fig. 2); this is between our SABER (~1.5 K) and reanalyses (~0.3 K) results (**Fig. 3a**). Swinbank et al. (1999) also analyzed MLS measurements (in 1992 only) and showed that the extratropical maxima in the upper stratosphere is 3-3.5 K in January; our analysis showed that it is >4 K for SABER and ~3 K in reanalyses in January (for 2006-2012 mean; not shown). Keckhut et al. (1996) reported that

UARS/MLS results are quite consistent with lidar measurements over a station in southern France (at 44°N). This latitude is close to the location of the amplitude maxima in the extratropical upper stratosphere we find for the $S_1$ migrating tide (**Fig. 1**). Huang et al. (2010) pointed out that the local upward phase progression between 35 and 60 km in SABER (c.f., **Fig. 3b**) is not observed in the measurements from the CRISTA during 5-11 November 1994 (Oberheide et al., 2000); the CRISTA results look similar to the present tidal determinations in the various reanalysis data sets. To summarize, it seems that there is

enough uncertainty concerning the $S_1$ migrating tide represented in the SABER data that further investigation may be needed before attributing the systematic differences we found between SABER and the reanalyses considered here.

The seasonal variation of amplitude of $S_1$ migrating tide averaged in the tropics (15°S-15°N) is shown in **Figure 6.** Monthly tides during 2006-2012 are calculated both for SABER and reanalyses; but for SABER, the results of each month are derived

from 60-day data (e.g., the results in January are from 15[th] December through 15 February). All data sets show that the amplitude maximizes in February-March and again in July-August-September, in the stratosphere and the lower mesosphere; this semiannual variation is consistent with previous studies (e.g., Mukhtarov et al., 2009; Huang et al., 2010; SA12). Such seasonality has been attributed to the anti-symmetric Hough mode strengthening due to the meridional gradient of the zonal-mean zonal wind in the tropics (McLandress 2002; Sakazaki et al., 2013b). In the extratropics, on the other hand, all data sets

show that the amplitude maximizes in local summer in the stratosphere (not shown), presumably due to the enhanced ozone heating in the summer hemisphere.

### 4.2 Semidiurnal ($S_2$) migrating tide

**Figures 7 and 8** show the latitude-altitude distribution of amplitude and phase, respectively, for $S_2$ migrating tides in

temperature. **Figure 9** compares the vertical profiles of $S_2$ amplitude and phase averaged over 15°S-15°N and 30°-45°N. The amplitude is largest in the tropics, showing a local maximum around at 40-45 km (up to ~1.2 K), i.e. close to the location of ozone heating maximum. In the tropics, the phase shows a slight upward progression below ~40 km (**Fig. 9b**), indicating that the energy propagates downward from the ozone heating layer. Above ~40 km, the phase is almost constant, at least in the tropics. The long vertical wavelength and the significant downward energy propagation from the stratosphere are consistent

with classical tidal theory for $S_2$ migrating tides (Chapman and Lindzen, 1970).

The vertical profiles of amplitude and phase are in good agreement among the data sets particularly below ~45 km, except that the ERA-Interim shows a smaller amplitude in the tropics (**Fig. 8a**). Above ~45 km, the phase diverges among the data



sets (**Figs. 9b and 9d**). In contrast to the $S_1$ migrating tide, there is no systematic difference between SABER and reanalyses in the upper stratosphere and the lower stratosphere but the amplitude in reanalyses is systematically smaller than that in SABER between 20 and 30 km in altitude (**Fig. 9a**). Note that the $S_2$ tides in the stratosphere have not been examined in detail except for some ground-based lidar measurements (e.g., Keckhut et al., 1996; Leblanc et al., 1999; Kopp et al., 2015);

our study for the first time demonstrated its meridional-vertical structure.

**Figure 10** shows the vertical profiles of amplitude and phase for the temperature projected onto the (2,2) mode ($\Theta_2^2$), i.e. the gravest symmetric $S_2$ Hough mode (see **Fig. A1** for the structure of this mode). That is, the $S_2$ migrating tide is approximated by $\tilde{T}_2^2(z)\Theta_2^2(\theta)\cos(2\omega(t - \tilde{\alpha}_2^2(z)))$ and the vertical profiles of $\tilde{T}_2^2(z)$ and $\tilde{\alpha}_2^2(z)$ are shown. Note that

classical tidal theory predicts that the $S_2$ tidal response should consist of only modes with vertical propagation, in contrast to $S_1$. The profiles of this mode are similar to the observed profiles over the tropics (**Figs. 9a and 9b**), meaning that this mode dominates the $S_2$ migrating tide over the tropics. All data sets show that the amplitude maximizes in the upper stratosphere, although the amplitude in ERA-Interim is again smaller than the other data sets. The phase is in good agreement among the data sets below 45 km, but it shows a difference above 45 km. As for $S_1$ migrating tides, the variance among reanalyses

becomes large in the upper stratosphere and the lower mesosphere even for such a large-scale structure.

**Figure 11** shows the month-altitude distribution of amplitude for $S_2$ migrating tide. Although the SABER results are noisy, all data sets basically show that the amplitude maximizes twice in December-January-February and in June-July-August in the upper stratosphere and the lower mesosphere. In the lower and middle stratosphere, by contrast, the amplitude minimizes

during June-July-August; notably, this is similar to the seasonality of surface pressure tides (e.g., Díaz-Argandoña et al., 2016; Hamilton and Sakazaki, 2017). Such seasonality in the stratosphere was reported earlier by Li et al. (2015) using the CFSR reanalysis.

### 4.3 Nonmigrating tides

**Figure 12** shows the longitude-altitude distribution of annual-mean nonmigrating temperature tides at 0000 UTC, as averaged between 10°S and 10°N. All data sets show clear gravity-wave patterns being excited and emanating from the two major continents, namely, Africa (10-40°E) and South America (80-40°W) and also indicate a somewhat weaker wave source from the Maritime Continent (90-150°E) Westward (eastward) tilting waves correspond to the westward (eastward) propagating waves which are clear in the western (eastern) hemisphere. **Figure 13** compares in detail the longitudinal

variations of nonmigrating tides at several pressure levels. We see that the longitudinal variations agree well among the data sets. There is no systematic difference between SABER and reanalyses. The biggest outliers are JRA-55C and JRA-55AMIP which seem to display somewhat larger amplitudes than the full reanalyses.



The gravity-wave pattern documented here is consistent with the finding by Sakazaki et al. (2015b), who analyzed data from a high-resolution GCM as well as SABER and COSMIC GPS radio occultation measurements. Although they noted that the amplitude in their model was significantly larger than that for SABER and COSMIC, the present results in reanalyses are

quantitatively consistent with those in SABER. The strength of these gravity-wave patterns are likely linked to the diurnal cycle in precipitation (latent heating) over the continents (Sakazaki et al., 2015b).

We note that averaging data between 10°S and 10°N as was done in **Figs. 12 and 13** only extracts the symmetric components with respect to the equator. Sakazaki et al. (2015a) showed that antisymmetric components near the equator (i.e.,

as revealed by taking the difference between the 10°S-0° average and the 0°-10°N average) have a clear zonally-uniform component (zonal wavenumber 0) as well as the gravity-wave patterns emanating from the continents (Sakazaki et al., 2015a). **Figure 14** shows our present results for the latitude-altitude structure of the annual-mean, zonal wavenumber 0 component (zonal-mean temperature anomaly from the daily mean) at 0000 UTC. As found by Kuroda and Chiba (1995) and Sakazaki et al. (2015a), the anti-symmetric structure with respect to the equator is dominant, with a vertical wavelength

of ~15 km and confined mainly to within about 15° of the equator.

  **Figure 15** shows the zonal wavenumber dependence for the $S_1$ (24 hr) harmonic of non-migrating tides for each symmetric and anti-symmetric component (see Section 3.1). All data sets show that zonal wavenumber 0 (so called D0; particularly for anti-symmetric components as seen in **Fig. 14**), westward zonal wavenumbers 5 and 2 (DW5 and DW2), and eastward zonal

wavenumber 3 (DE3) are dominant, being consistent with previous studies (Forbes and Wu., 2006; Zhang et al., 2006; Sakazaki et al., 2015b). Particularly, DW5 in the stratosphere corresponds to the clear westward tilting waves in **Fig. 12** (Sakazaki et al., 2015b). Although the dominant wavenumbers agree among the data sets, their magnitudes display some differences. A marked difference is seen for DE3; the MERRA and MERRA-2 results are close to the SABER but the other reanalyses have larger amplitudes than SABER above the middle stratosphere (pressures less than 3 hPa).

  Sakazaki et al. (2015b) in their study of nonmigrating tides found that the westward-propagating waves from the continents penetrate deeply into the mesosphere during equinox but they are dissipated near the stratopause around solstice season likely due to filtering by zonal wind associated with the stratospheric semiannual oscillation (SAO). In the present project we confirmed that such features are discernable in all reanalysis data sets (not shown). For the zonally uniform pattern discussed

above, Sakazaki et al. (2015a) showed that it is most clear in June-July-August; this was also confirmed in all data sets in the present study (not shown).



## 5 Interannual Variations and Long-term Trends in Reanalysis representation of Tides

This section examines the interannual variations and long-term changes in $S_1$ and $S_2$ migrating tides as represented in the various reanalyses over the extended 1980-2012 period. **Figure 16a-c** shows the monthly amplitude of $S_1$ migrating tide averaged over 10°S-10°N at selected pressure levels in the stratosphere and the lower mesosphere (0.4 hPa, 3 hPa and 10

hPa). The seasonal variations have been removed by applying a 12-month running mean. First, all reanalyses show similar inter-annual variations with a peak-to-peak difference of up to 0.5 K. The time series of two quasi-biennial oscillation (QBO) indices, the zonal wind at 10 hPa and 30 hPa over Singapore after the deseasonalization (12-month running mean) and normalization, are shown in **Figure 16d**. It is clear that the main interannual variations in tides are synchronized with the QBO cycle in stratospheric zonal wind. The modulation of $S_1$ tides by tropical stratospheric QBO in mean winds has been

reported in satellite measurements from the stratosphere through the MLT (e.g., Burrage et al., 1995; Mukhtarov et al., 2009). The QBO in zonal wind itself is represented quite well in the reanalyses considered here (including JRA-55C) (Konayashi et al., 2014; Kawatani et al., 2016). Note that the free-running JRA55-AMIP model does not generate a QBO in the tropical stratospheric mean circulation (Kobayashi et al., 2014) and correspondingly there is no QBO apparent in the $S_1$ tidal amplitudes (Fig. 16b,c).

The difference in tidal amplitudes among the reanalyses depends both on vertical level and it changes through the full period. In the lower mesosphere at 0.4 hPa (**Fig. 16a**), the amplitudes for MERRA and MERRA-2 are larger than that for ERA-Interim and JRA-55. This pattern continues for the three decades except that the MERRA-2 amplitude became smaller after ~2004, likely corresponding to the assimilation of MLS temperature starting in 2004. Since no other measurements are

assimilated in the lower mesosphere, the reanalyses in this altitude region are presumably strongly dependent on the tides simulated in forecast model used in producing each reanalysis. **Fig. 18a** shows the variance in amplitude of $S_1$ migrating tides averaged over 10°S-10°N among the four reanalyses, MERRA-2, MERRA, JRA-55, ERA-Interim (CFSR is not included because it because its CDAS-T382 integration has ended in December 2010) plotted as a function of altitude and time. In the lower mesosphere the variance among the reanalysis data sets is large (~1 K) and fairly steady throughout the

entire record..

In the upper stratosphere at 3 hPa, it is clear that the variance among the reanalyses was much larger before 2000 than after 2000 (**Figs. 16b and 18a**). Notably, the amplitude in JRA-55 increases abruptly in ~2000 to approach to the results of other reanalyses, while the JRA55-C does not show any systematic changes even after ~2000. This clearly indicates that the

satellite observations, which are assimilated for JRA-55 but not for JRA-55C, are responsible for the drastic improvement around 2000. Actually, the years around 2000 corresponds to the timing of the TOVS-to-ATOVS transition. ATOVS has the AMSU, which has more channels in the upper stratosphere with narrower weighting functions compared to the SSU on TOVS, so that the representation of the stratospheric dynamical fields significantly improved at this time. For JRA-55, SSU



was assimilated until ~2000 while AMSU started to be assimilated in ~1999 (both SSU and AMSU were assimilated during 1999-2000; see Figure 8 of Fujiwara et al., 2017). Artificial jumps around 2000 have been reported for other features of the circulation in the reanalysis data sets such as climatological temperature (Long et al., 2017) and the zonal wind in the tropical stratosphere (Kawatani et al., 2016).

Finally in the middle-lower stratosphere at 10 hPa, the variance is relatively small for the entire period compared to that at higher vertical levels; but as at the 3 hPa level, an abrupt decrease of variance is observed after ~2000 (**Figs. 16c, 18a**). In 1990s, quite large amplitudes are sometimes observed in ERA-Interim.

10 **Figure 17** shows the monthly amplitude of the $S_2$ migrating tide averaged over 10°S-10°N, while the variance in this quantity among the four reanalyses is shown in **Fig. 18b**. The QBO-related variation observed for $S_1$ migrating tides (**Fig. 16**) is not clear for the $S_2$ tide. An abrupt change due to the TOVS-to-ATOVS transition around 2000 does not seem clear for the $S_2$ tidal amplitudes, expect possibly for the CFSR data set which is a strong outlier at 10 hPa before ~1999 and somewhat more consistent with the other reanalyses after ~1999 (**Fig. 17c**) (for CFSR, SSU was assimilated until ~1998; see Fig. 8 of 15 Fujiwara et al., 2017). However, other strange interannual variations are observed particularly before 2000; notably, the ERA-Interim shows a 'saw-tooth'-pattern of changes at 0.4 hPa and 3 hPa until ~2000. This likely is caused by the orbital drift of TOVS and the transition between different NOAA satellites carrying the TOVS (e.g., Zou et al., 2014). For example, TOVS was onboard NOAA-9 between 1985 and 1988 and was onboard NOAA-11 between 1988 and 1994; the orbital drift of NOAA-9 (NOAA-11) itself likely corresponds to the gradual increase in $S_2$ amplitude over 1985-1988 (1988-1994), and 20 the transition between the two satellites likely corresponds to the abrupt reduction seen in ERA-Interim representation of $S_2$ amplitude in 1988 at 0.4 hPa level (**Fig 17a**).

## 6 Summary and Discussion

This study investigated the solar tides seen in the temperature in the stratosphere and the lower mesosphere using state-of-the-art reanalysis data sets included in the S-RIP intercomparison project and comparing with independent SABER 25 measurements during 2006-2012. Diurnal ($S_1$) migrating tides, semidiurnal ($S_2$) migrating tides, and nonmigrating tides are extracted and discussed individually. Overall, the reanalysis results are found to be quite consistent with those from SABER in a qualitative way, such as the three-dimensional structure, dominant wavenumbers (for nonmigrating tides) and their seasonality. The spread among the reanalyses also increases with altitude and is fairly large in the lower mesosphere where few actual observations are assimilated, leaving the reanalysis fields dependent on the tides simulated in the forecast model 30 used in each reanalysis procedure.



A marked systematic difference between SABER and reanalyses is seen for the amplitude and phase profiles for $S_1$ migrating tides above 40 km. S12 noticed this issue using MERRA, ERA-Interim, CFSR and JRA-25 but this study confirmed such difference for the more recent reanalyses (MERRA-2 and JRA-55) as well. Swinbank et al. (1999) found that the assimilation of SSU measurements damps the representation of the tidal amplitude in a reanalysis in the upper stratosphere.

The comparison of JRA-55 family data sets in our study, however, suggests that the assimilation does not degrade tides at least in the present day (i.e., 2006-2012 period) and in the JRA-55 system. A Hough mode decomposition further showed that such SABER-reanalyses difference can be attributed primarily to the amplitude of the trapped (1,-1) mode response in the stratosphere. This could be explained if either the stratospheric ozone heating is underestimated in the forecast models used to produce the reanalyses, or, SABER temperatures have some systematic local time biases. We also compared the

vertical profile of ascending-descending temperature differences from Aura/MLS measurements, which is a good indicator of magnitude of $S_1$ migrating tides, to reanalysis temperatures sampled at the same local times. Our results suggest that the $S_1$ tide in the reanalyses are closer to those derived from Aura/MLS than SABER observations. An inter-comparison with available ground-based measurements may be helpful to resolve this issue.

The evolution of tidal amplitudes derived from the reanalyses over the extended 1980-2012 period shows a clear QBO signals except for JRA-55AMIP data which has no QBO in equatorial stratospheric mean circulation. On the other hand, it is suggested that any long term changes are primarily artificial and are driven by several changes of input data employed. The largest impact is caused by the TOVS-to-ATOVS transition, as well as the changes of NOAA satellites carrying TOVS. The tides as represented in MERRA-2 reanalyses are also affected by the incorporation of MLS data starting in 2004. How much

influence these changes have on tides depend on each reanalysis system and also on the tidal frequency (i.e., $S_1$ or $S_2$). This finding indicates that the inter-comparison results depend on the analysis period and artificial discontinuities in the data stream that were assimilated make it quite difficult to detect natural long-term trends of the tides in the middle atmosphere.

Some current global atmospheric models cover the region from the surface up to the MLT (often refered to as "whole

atmosphere models"). Such models are sometimes integrated with several dynamical variables nudged toward reanalysis data in order to reproduce the realistic day-to-day variations in the upper atmosphere that are often connected to tidal variations (e.g., Jin et al., 2012; Pedatella et al., 2014). In this respect, tides in reanalysis data provide an important 'lower boundary condition' for simulations of upper-air dynamics. Tides in reanalyses are also used for correcting the diurnal anomaly or drift seen in Sun-synchronous satellite measurements. Zou et al. (2014) corrected the local-time drift in SSU

temperature measurements by using temperature tides in MERRA. The present evaluation of stratospheric tides thus should be helpful for estimating the uncertainty associated with using reanalyses for such applications.



**Data Availability**

Diurnal monthly reanalysis data sets are publicly available, as follows:

1) MERRA-2: https://disc.sci.gsfc.nasa.gov/uui/datasets?keywords=%22MERRA-2%22 (doi: 10.5067/6EGRBNEBMIYS)

2) MERRA: https://disc.sci.gsfc.nasa.gov/uui/datasets?keywords=%22MERRA%2225 (doi: 10.5067/BUFAR1DPYIR9)

3) JRA-55: Through the DIAS at http://search.diasjp.net/en/dataset/JRA55

4) JRA-55C: Through the DIAS at http://search.diasjp.net/en/dataset/JRA55_C

5) JRA-55AMIP: Through NCAR RDA at https://doi.org/10.5065/D6T72FHN

6) ERA-I: http://apps.ecmwf.int/datasets/

7) CFSR: Through NCAR RDA https://doi.org/10.5065/D6DN438J

SABER data can be downloaded from the ftp site at ftp://saber.gats-inc.com/custom/Temp_O3/. MLS data can be downloaded from https://disc.gsfc.nasa.gov/datasets/ML2T_V003/summary.

**Appendix A: Major abbreviations and terms**

AMSU  Advanced Microwave Sounding Unit

Aura  A satellite in the EOS A-Train satellite constellation

ATOVS  Advanced TIROS Operational Vertical Sounder

CFSR  Climate Forecast System Reanalysis of NCEP

COSMIC  Constellation Observing System for Meteorology, Ionosphere, and Climate

CRISTA  CRyogenic Infrared Spectrometers and Telescopes for the Atmosphere

ECMWF  European Centre for Medium-Range Weather Forecasts

ERA-Interim  ECMWF interim reanalysis

JCDAS  JMA Climate Data Assimilation System

JRA-25  Japanese 25-year Reanalysis

JRA-55  Japanese 55-year Reanalysis

JRA-55AMIP  Japanese 55-year Reanalysis based on AMIP-type simulations

JRA-55C  Japanese 55-year Reanalysis assimilating Conventional observations only

MERRA  Modern Era Retrospective-Analysis for Research

MLS  Microwave Limb Sounder

NCAR  National Center for Atmospheric Research

NCEP  National Centers for Environmental Prediction of NOAA

NOAA  National Oceanic and Atmospheric Administration

SABER  Sounding of the Atmosphere using Broadband Emission Radiometry

SPARC  Stratosphere-troposphere Processes And their Role in Climate

SSU  Stratospheric Sounding Unit

TIMED  Thermosphere-Ionosphere-Mesosphere-Energetics and Dynamics

UARS  Upper Atmosphere Research Satellite

**Acknowledgements**

We are grateful to Kevin Hamilton for valuable comments and suggestions on the original manuscript. We thank Yoko Naito for processing the original MLS ver. 3.3 data, and Chiaki Kobayashi and Yayoi Harada for helpful discussions on the JRA-55 results. We also thank NASA's GMAO, ECMWF, JMA, and NCEP for providing reanalysis data sets. This study was in part supported by the Japan Society for the Promotion of Science (JSPS) through Grants-in-Aid for Scientific Research (15K17761 and 16K05548). Figures were produced using the GFD-DENNOU Library. The DIAS dataset is archived and provided under the framework of the Data Integration and Analysis System (DIAS) funded by the Japan Ministry of Education, Culture, Sports, Science and Technology (MEXT).

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





**Table 1**: List of reanalyses used in this study. Also included is the JRA-55AMIP data set which represents results of a free running version of the forecast model used to produce the JRA55 and JRA-55C reanalyses.

| (Reanalysis) | Time resolution of output | Output top | Model Top | Reference |
|---|---|---|---|---|
| **MERRA-2** | 3-hourly | 0.1 hPa | 0.01 hPa | GMAO (2015), Gelaro et al. (2017) |
| **MERRA** | 3-hourly | 0.1 hPa | 0.01 hPa | Rienecker et al. (2011) |
| **JRA-55** | 6-hourly | 0.1 hPa | 0.1 hPa | Kobayashi et al. (2015) |
| **JRA-55C** | 6-hourly | 0.1 hPa | 0.1 hPa | Kobayashi et al. (2014) |
| **JRA-55AMIP** | 6-hourly | 1 hPa | 0.1 hPa | |
| **ERA-Interim** | 6-hourly | 0.1 hPa | 0.1 hPa | Dee et al. (2011) |
| **CFSR** | 6-hourly | 1 hPa | 0.266 hPa | Saha et al. (2010) |




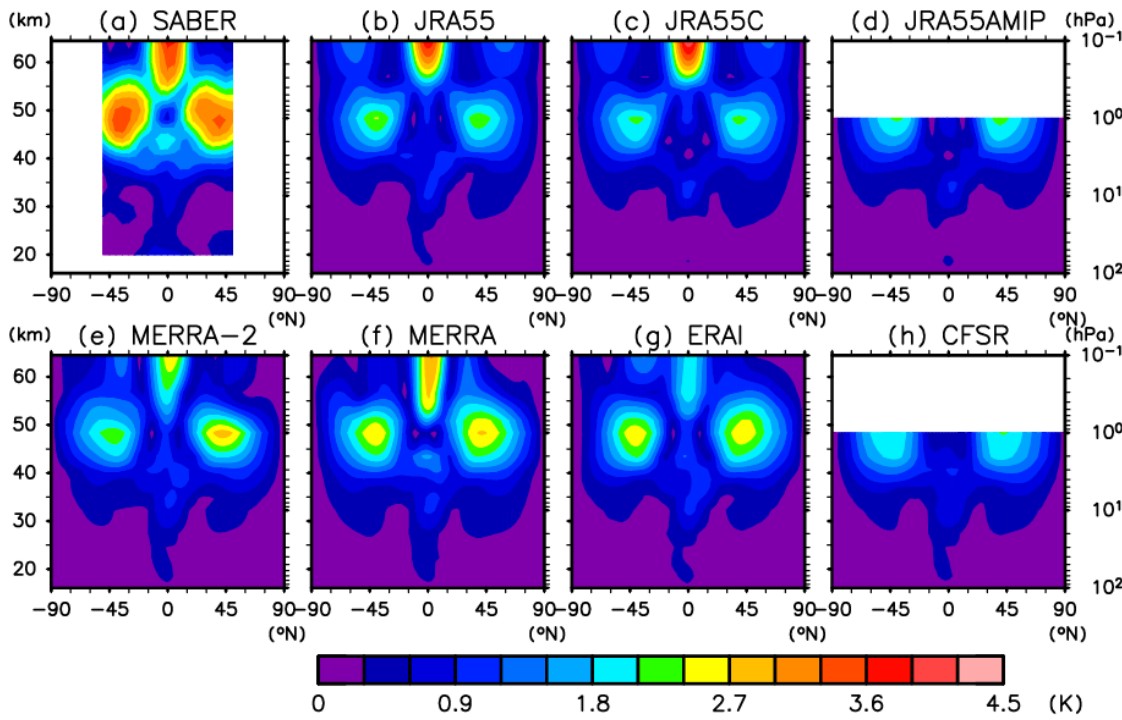

**Figure 1:** Latitude-altitude distribution of amplitude for diurnal ($S_1$) migrating tide in temperature, as derived from (a) SABER, (b) JRA55, (c) JRA55-C, (d) JRA55-AMIP, (d) MERRA-2, (e) MERRA, (f) ERA-Interim, and (h) CFSR.





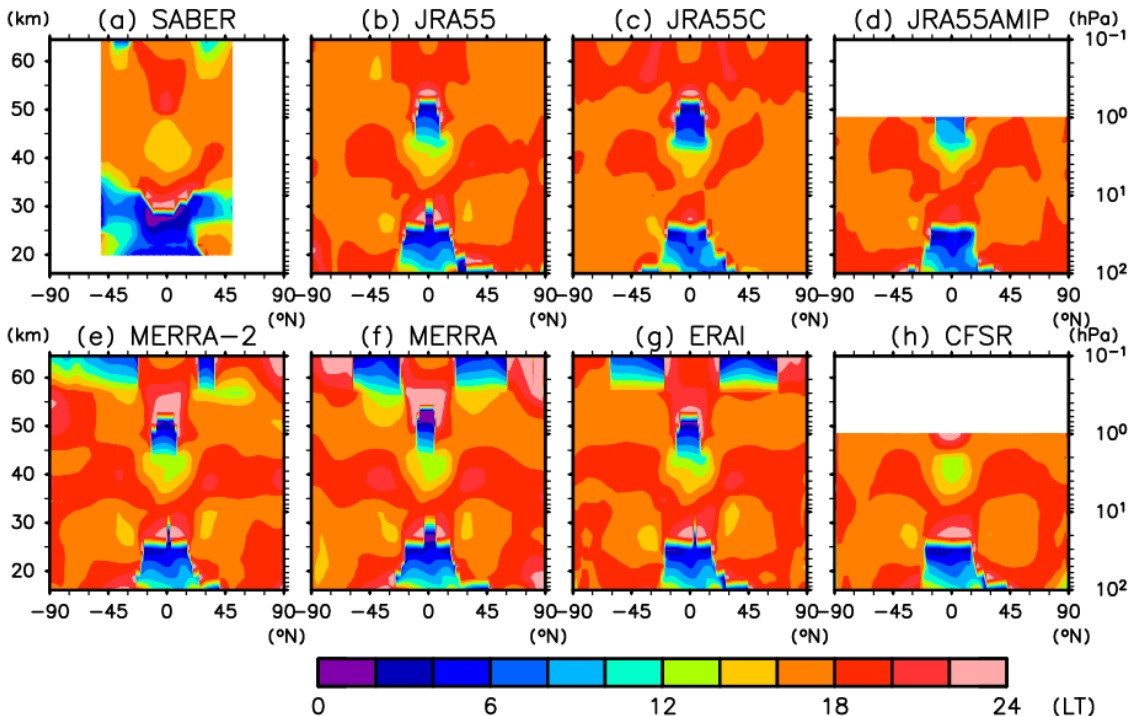

**Figure 2:** As is Fig. 1 but for phase (LT at which the $S_1$ temperature variation is maximum).




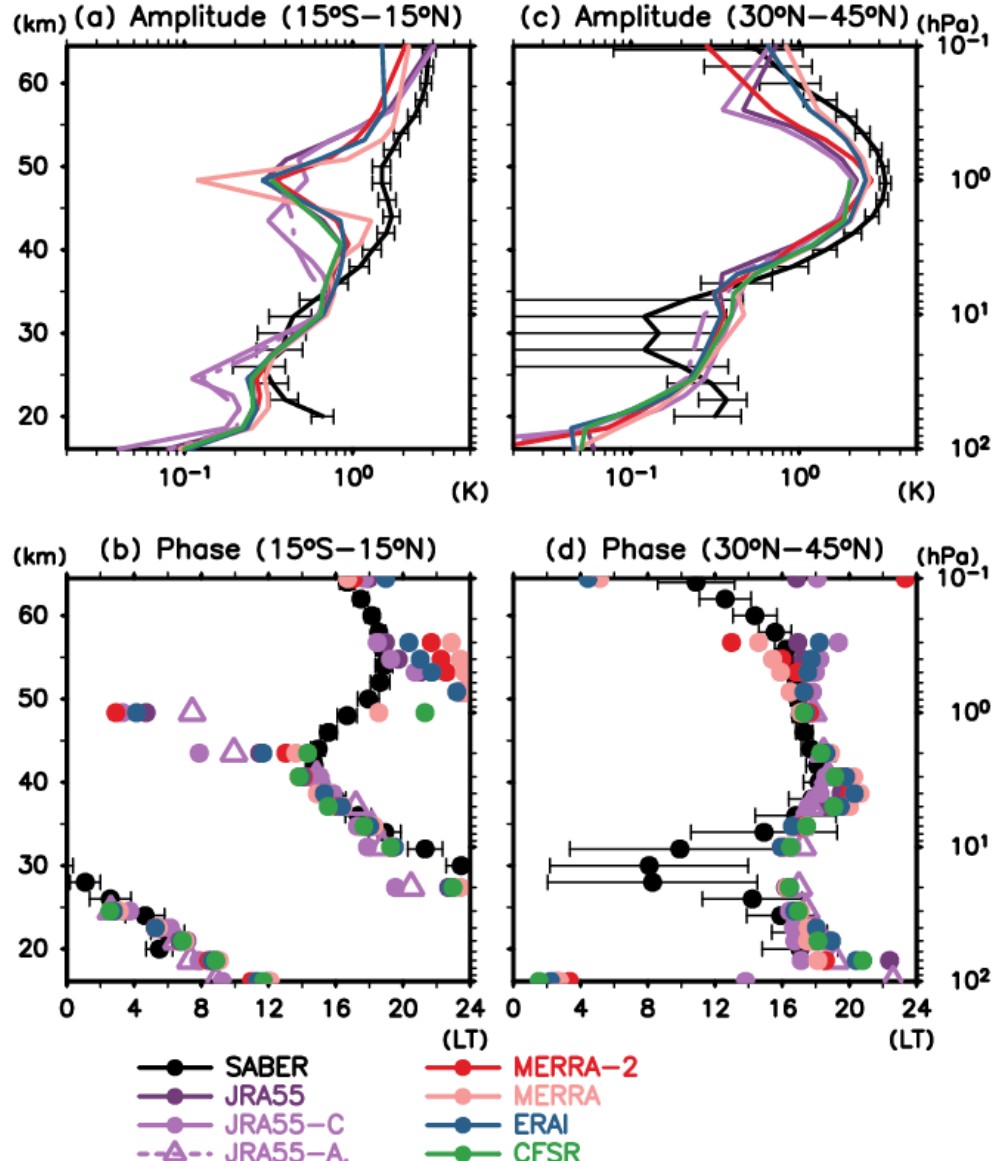

**Figure 3**: Vertical profile of (a, c) amplitude and (b, d) phase of the diurnal ($S_1$) migrating tide averaged for (a-b) 15°S-15°N and (c-d) 30°N-45°N, derived from different datasets. Horizontal bars show 95% confidential levels in a $t$ test for SABER results. For the statistical test, the error is defined as the 95% confidential level for daily anomaly (composite value) at each hourly universal time; this quantity has been propagated to the error of amplitude and phase for diurnal migrating tides following the error propagation theory.





**Figure 4**: Vertical profile of (a, c) amplitude and (b, d) phase of (a-b) the first propagating Hough mode and (c-d) the first trapped Hough mode, for the diurnal ($S_1$) migrating tide.



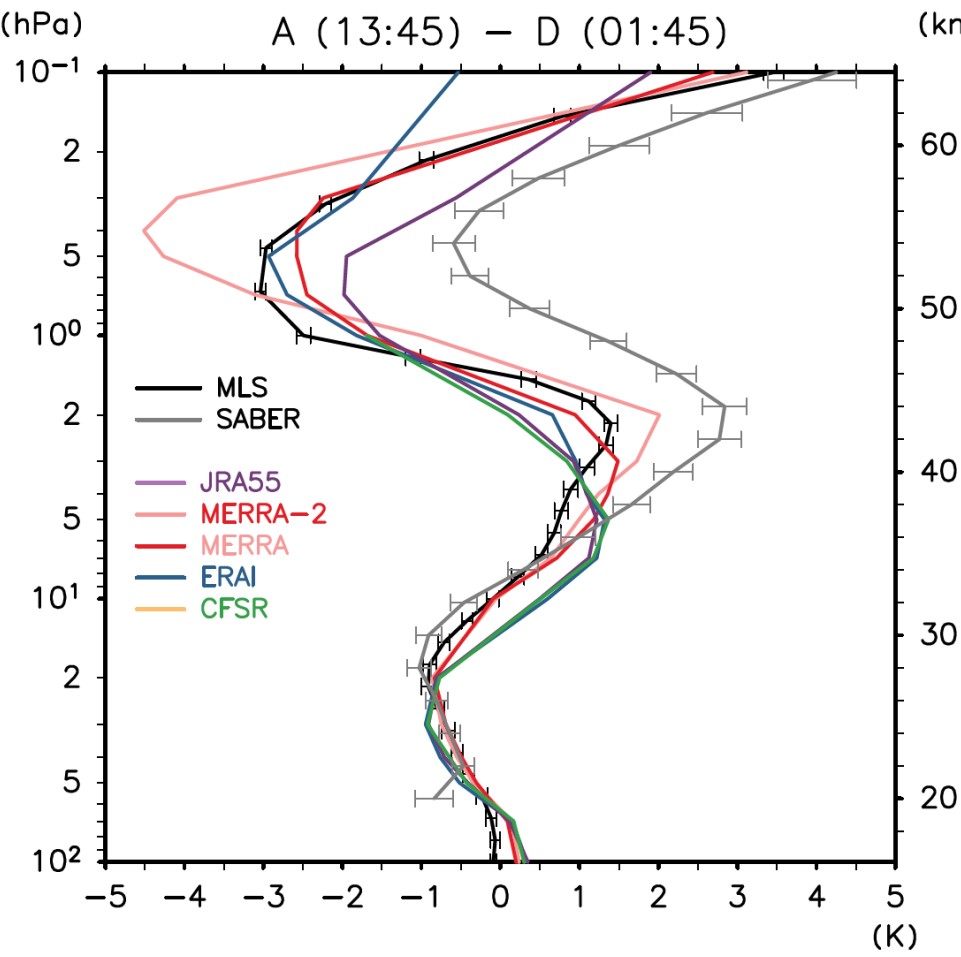

**Figure 5**: Vertical profile of difference between ascending and descending profiles of temperature in MLS measurements compared to the 1345 minus 0145 LT temperature difference sampled from the SABER data and various reanalysis data sets. Horizontal bars show 95% confidential levels with t test.  Results are annual means for the 7 year period 2006-2012.



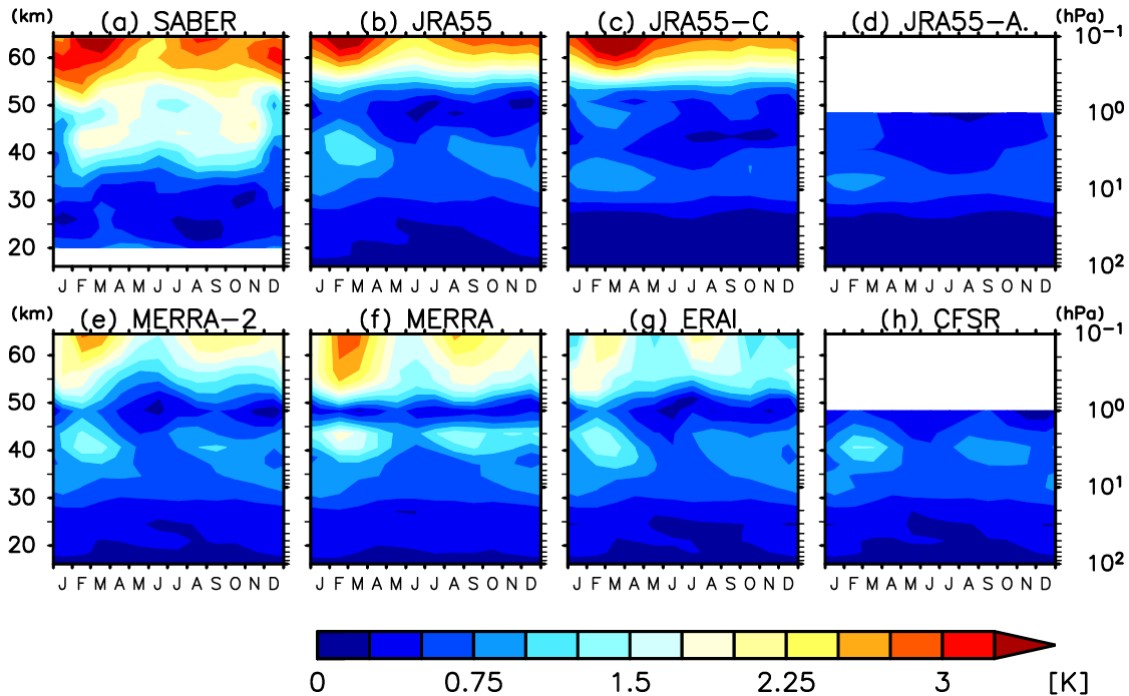

**Figure 6**: Month-altitude distribution of amplitude of diurnal ($S_1$) migrating tides averaged between 15°S and 15°N from (a) SABER, (b) JRA-55, (c) JRA-55C, (d) JRA-55AMIP, (e) MERRA-2, (f) MERRA, (g) ERA-Interim and (h) CFSR.




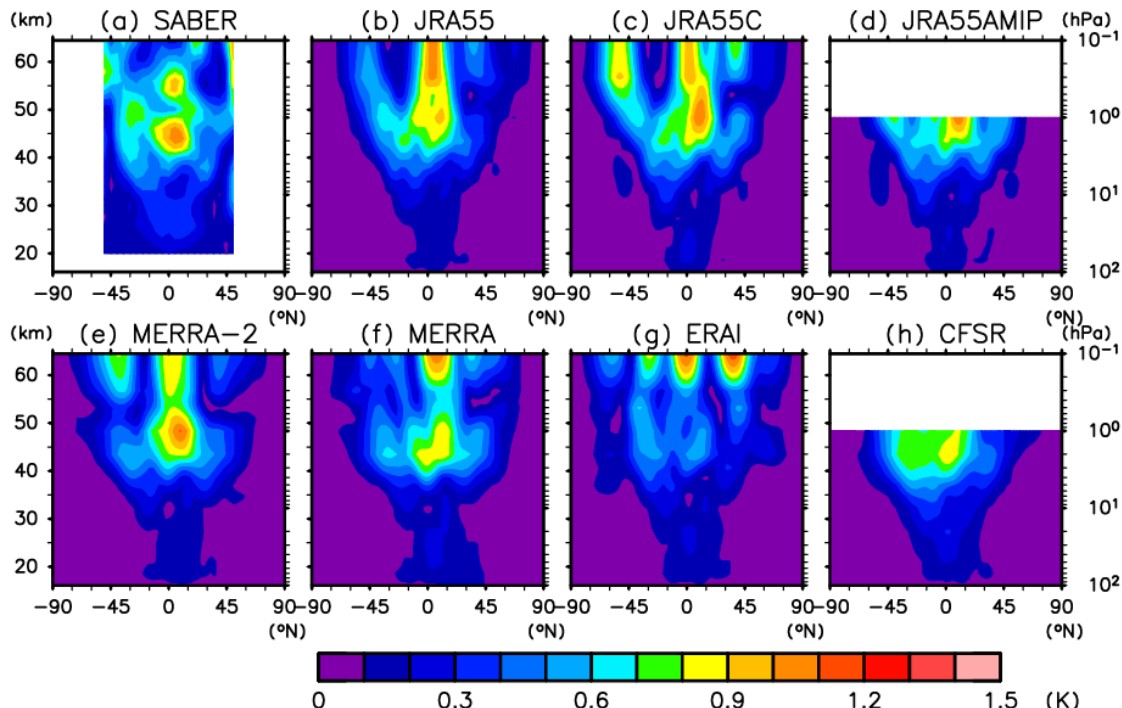

**Figure 7**: As is Fig. 1 but for semidiurnal ($S_2$) migrating tide.





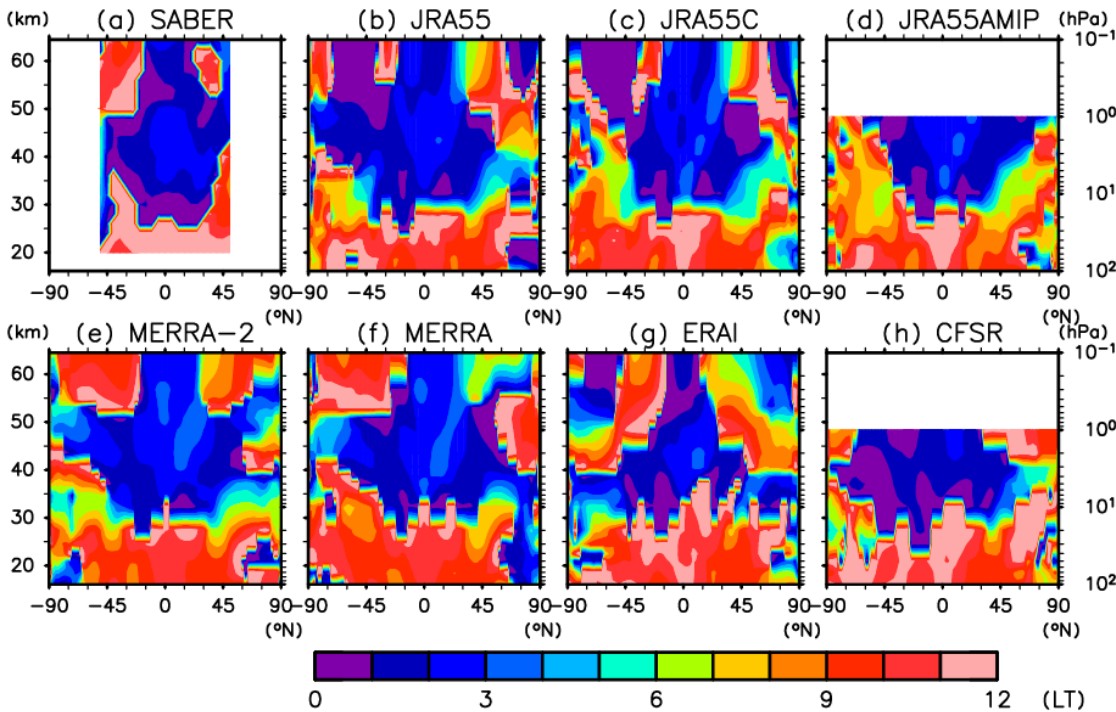

**Figure 8**: As is Fig. 2 but for the semidiurnal ($S_2$) migrating tide.



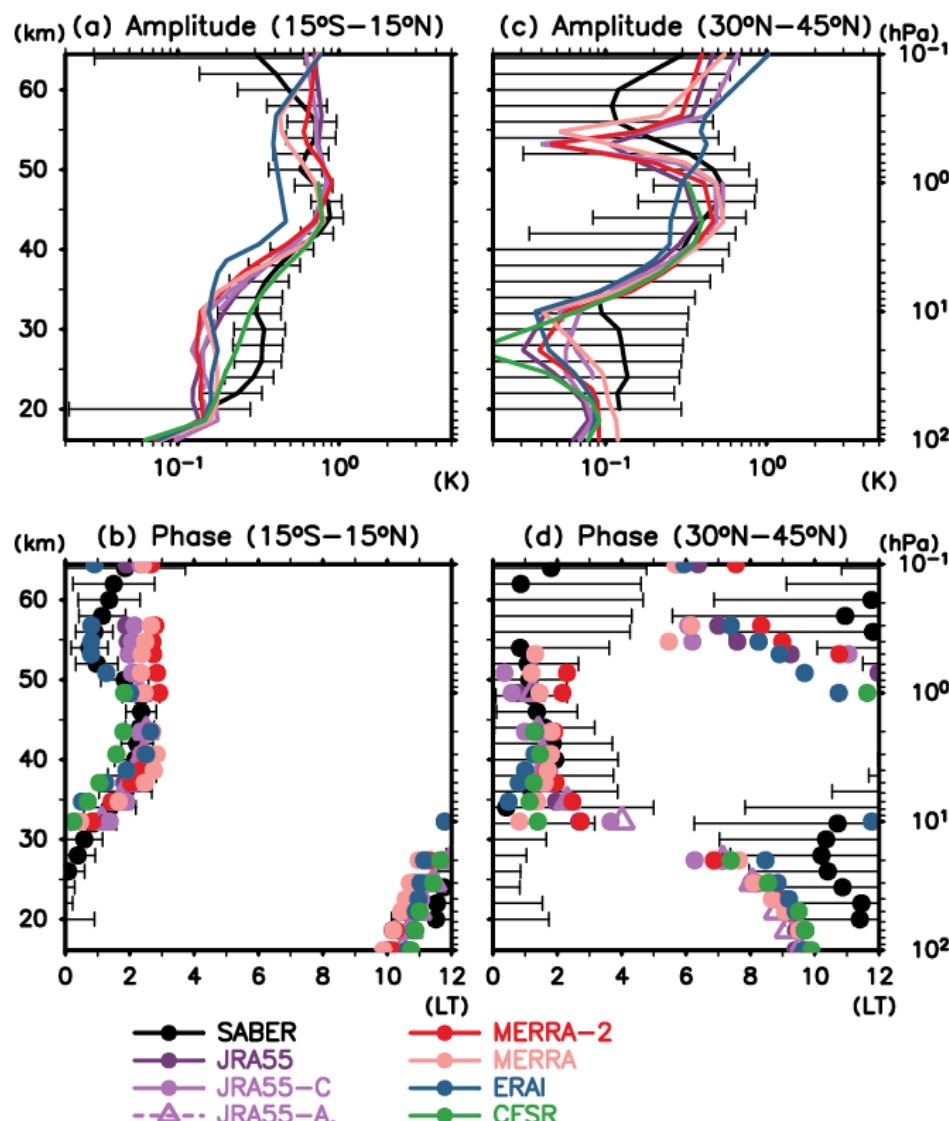

**Figure 9**: As is Fig. 3 but for the semidiurnal ($S_2$) migrating tide.





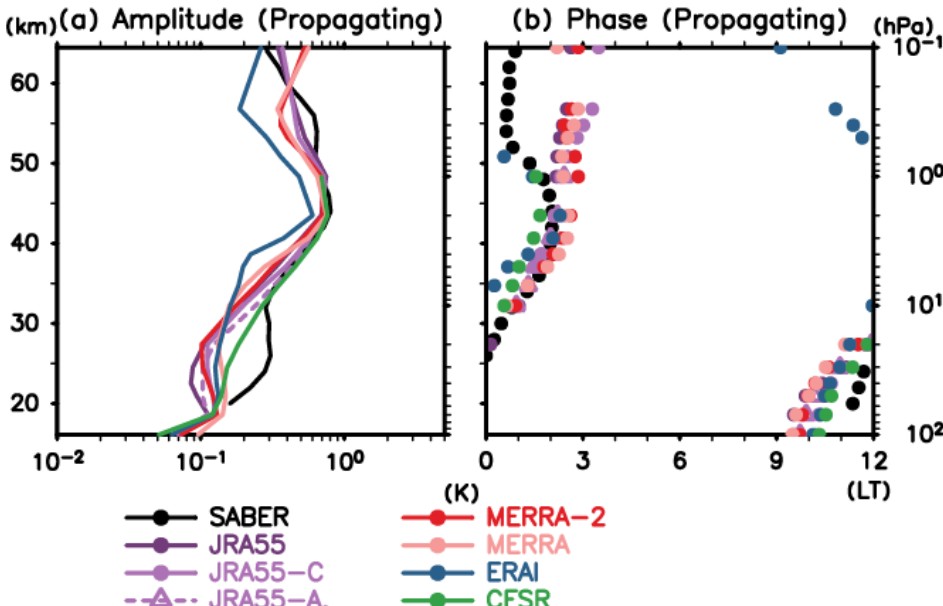

**Fig. 10**: As is Fig. 4 but for the gravest symmetric Hough mode (2,2) of the semidiurnal ($S_2$) migrating tide.



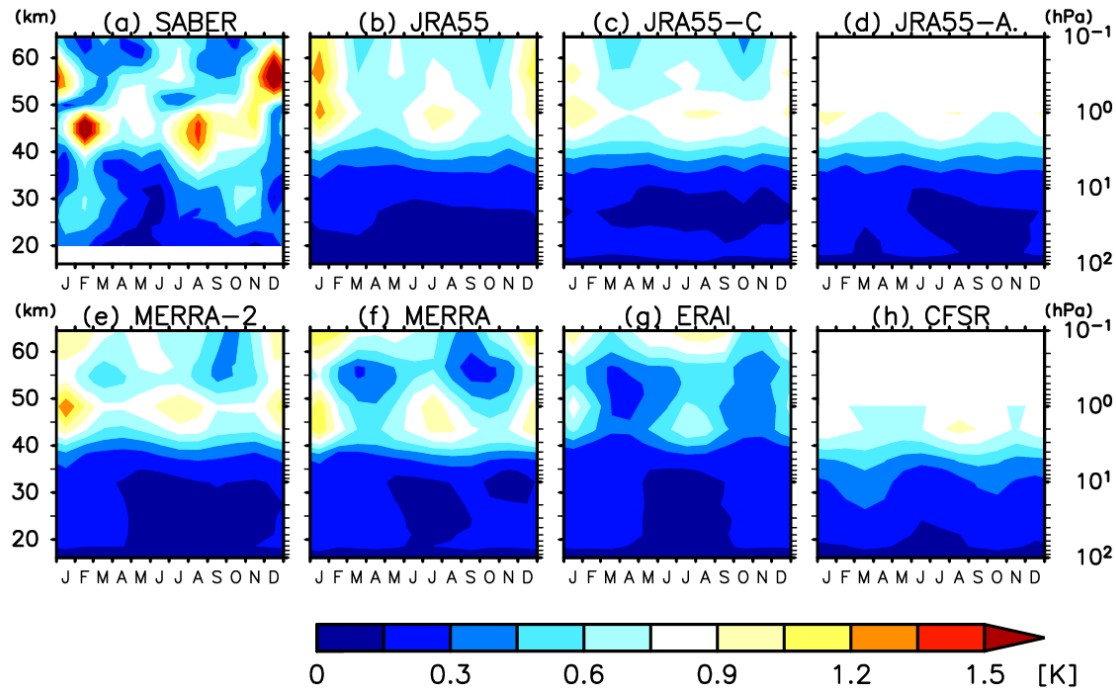

**Figure 11**: As is Fig. 6 but for the semidiurnal ($S_2$) migrating tide.





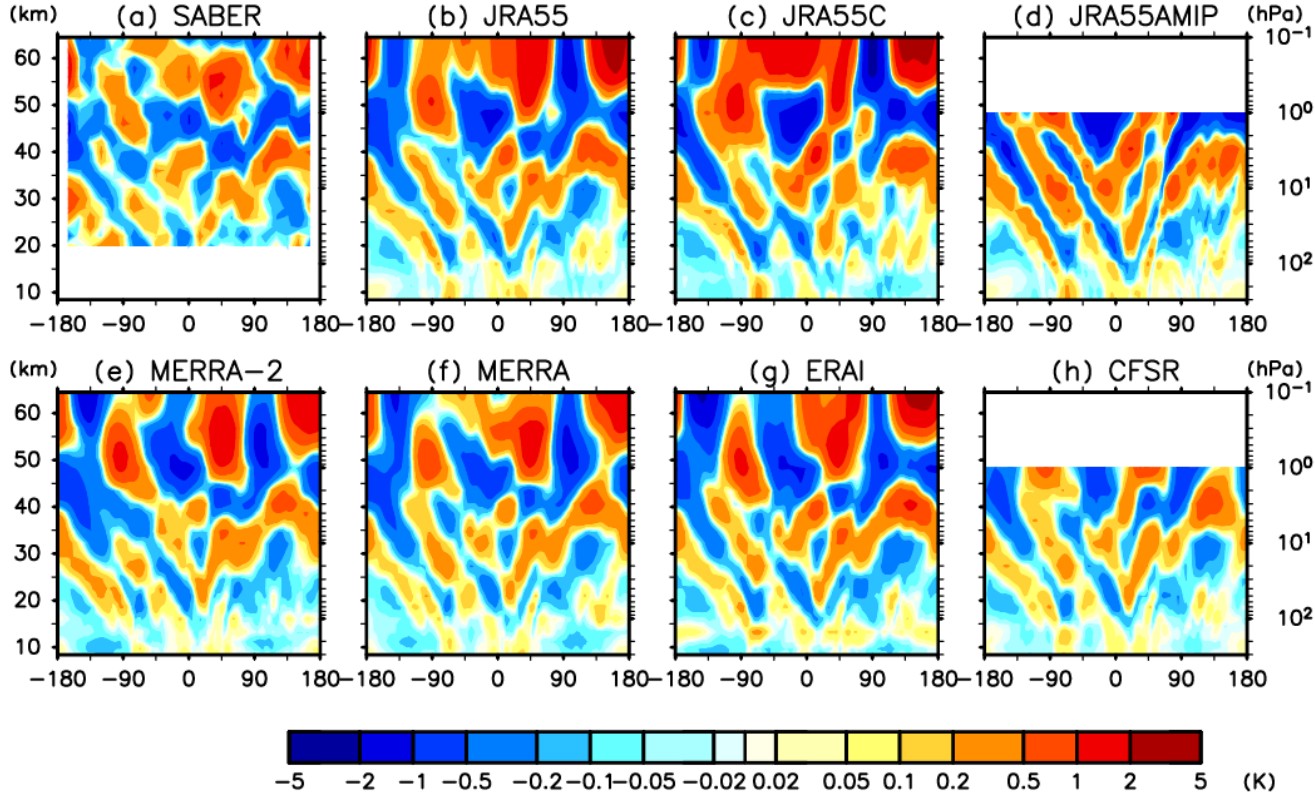

**Figure 12**: Longitude-altitude distribution of annual-mean nonmigrating temperature tides at 0000 UTC, as derived from (a) SABER, (b) JRA55, (c) JRA55-C, (d) JRA55-AMIP, (e) MERRA-2, (f) MERRA, (g) ERA-Interim, and (h) CFSR.




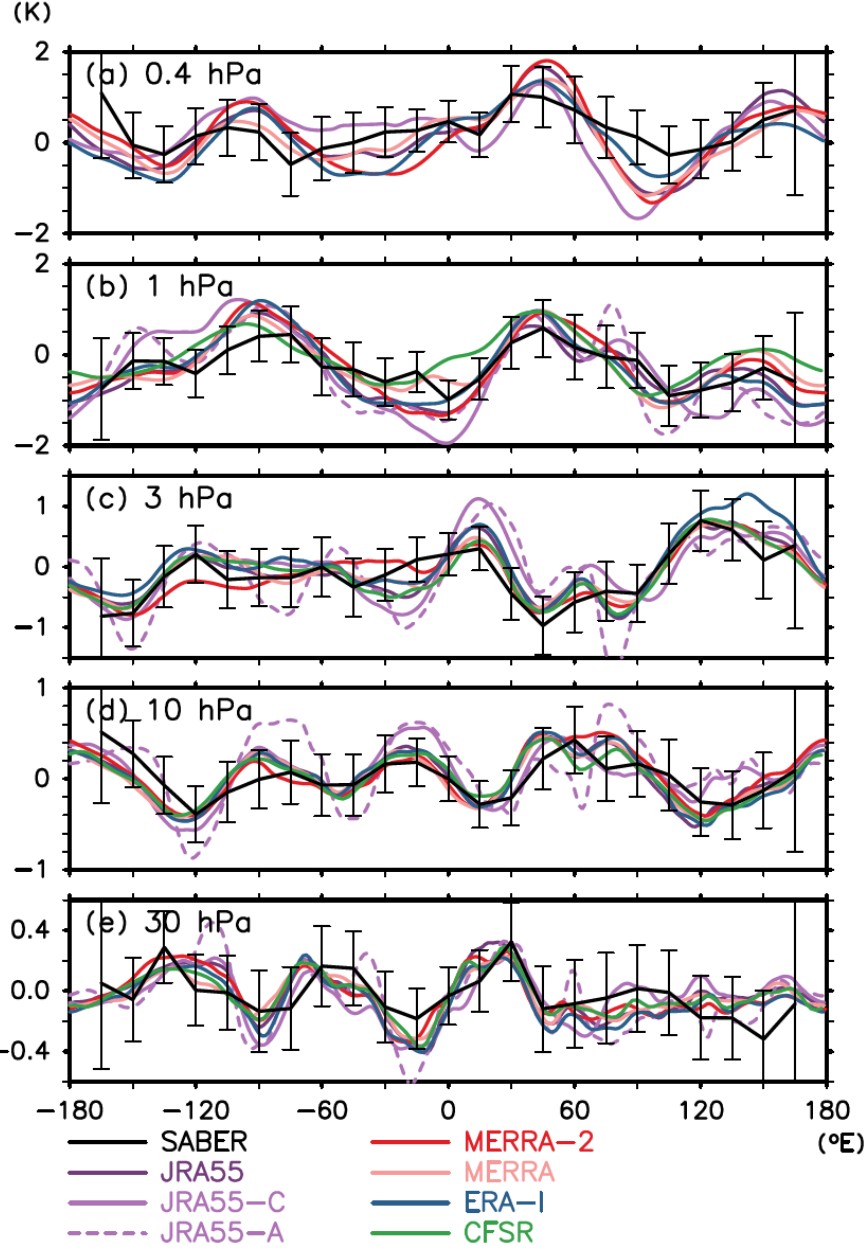

**Figure 13:** Longitudinal variation of nonmigrating tides at 0000 UTC averaged between 10°S and 10°N, at (a) 0.4 hPa, (b) 1 hPa, (c) 3 hPa, (d) 10 hPa and (e) 30 hPa. Vertical bars show 95% confidence level estimated by a *t* test.



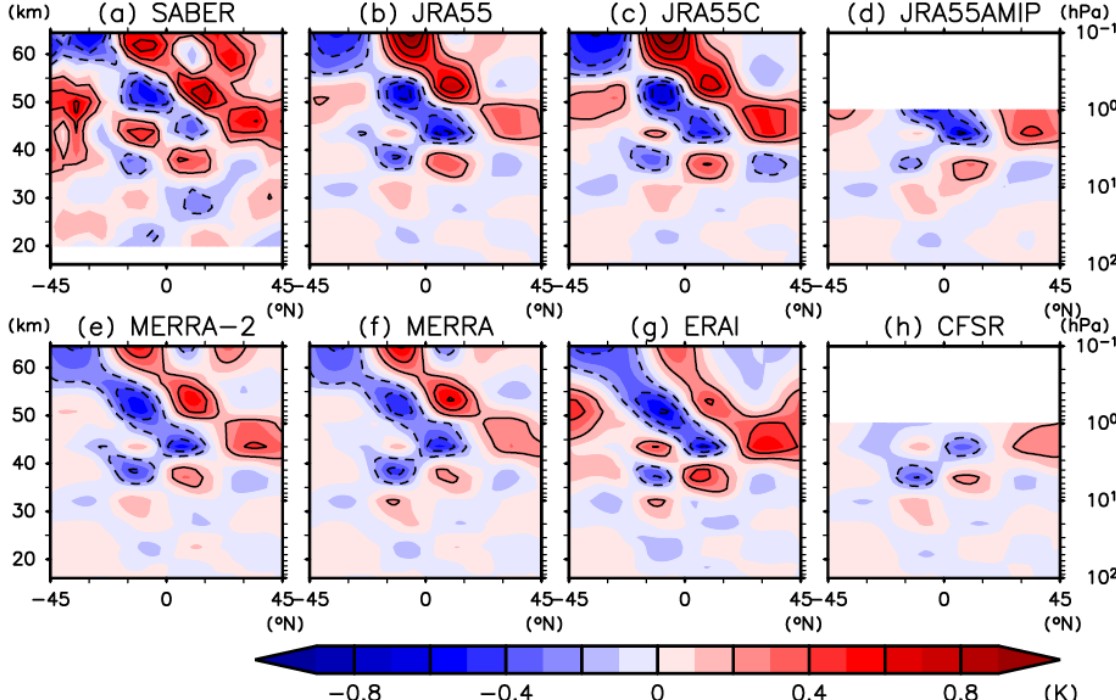

**Figure 14:** Latitude-altitude distribution of zonally-uniform nonmigrating tides (zonal wavenumber 0 component) at 0000 UTC, as derived from (a) SABER, (b) JRA55, (c) JRA55-C, (d) JRA55-AMIP, (e) MERRA-2, (f) MERRA, (g) ERA-Interim, and (h) CFSR. Contour interval is 0.2 K.


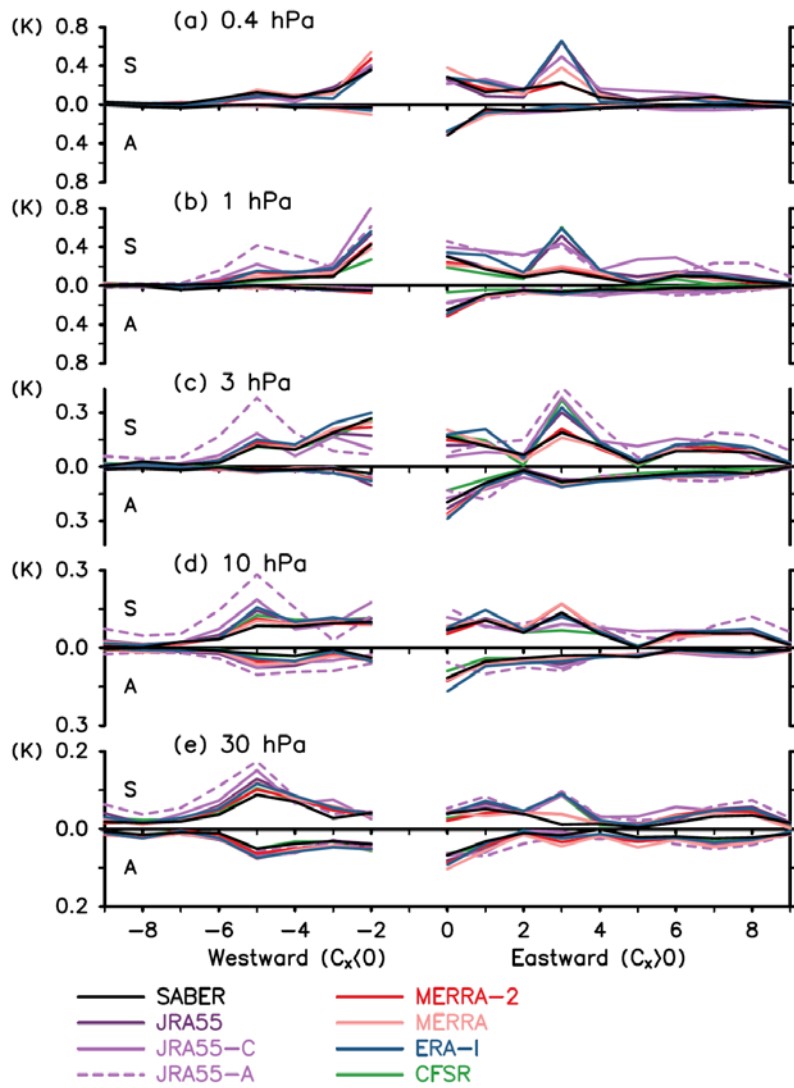

**Figure 15**: Amplitudes for each zonal wavenumber component of diurnal ($S_1$) nonmigrating tides for the region between 10°S and 10°N, at (a) 0.4 hPa, (b) 1 hPa, (c) 3 hPa, (d) 10 hPa and (e) 30 hPa. Top and bottom half in each panel shows the results of symmetric and anti-

5  symmetric components, respectively. Positive and negative wavenumbers are for the eastward and westward travelling waves, respectively. The $S_1$ migrating tide ($s = -1$) is not shown.





**Figure 16**: (a-c) Long-term changes in amplitude of the diurnal ($S_1$) migrating tide averaged between 10°S and 10°N after applying a 12-month moving average, at (a) 0.4 hPa, (b) 3 hPa and (c) 10 hPa, as derived from reanalyses.(d) Two QBO indices defined as the deseasonalized (12-month moving average), normalized zonal wind over Singapore at (solid gray curve) 10 hPa and (dashed gray curve) 30 hPa.







**Figure 17**: As is Fig. 16a-c but for the semidiurnal ($S_2$) migrating tide.





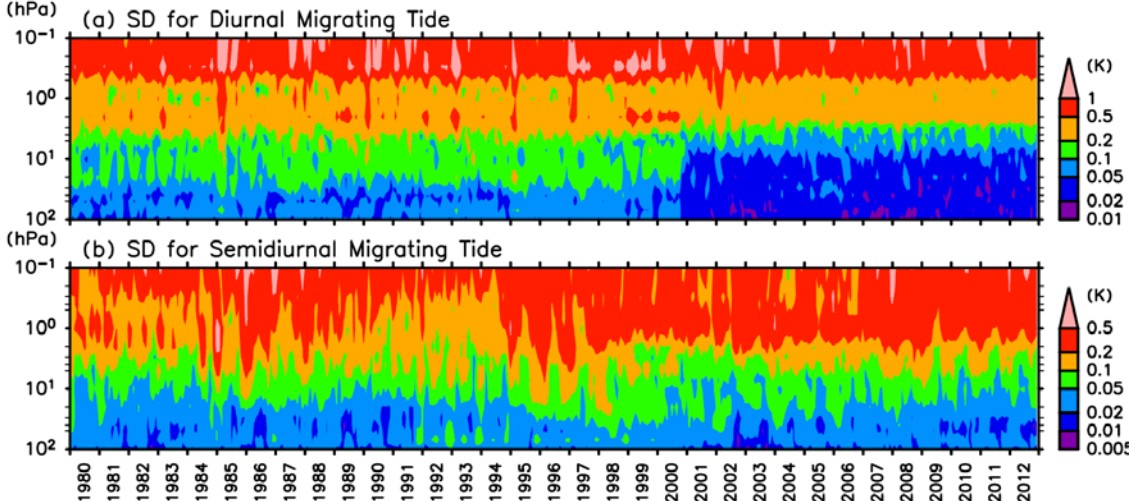

**Figure 18**: Time-altitude distributions of standard deviation among the four reanalyses (MERRA, MERRA-2, ERA-Interim and JRA55) for (a) amplitude of the diurnal ($S_1$) migrating tide and (b) the amplitude of the semidiurnal ($S_2$) migrating tide, averaged over 10°S-10°N.





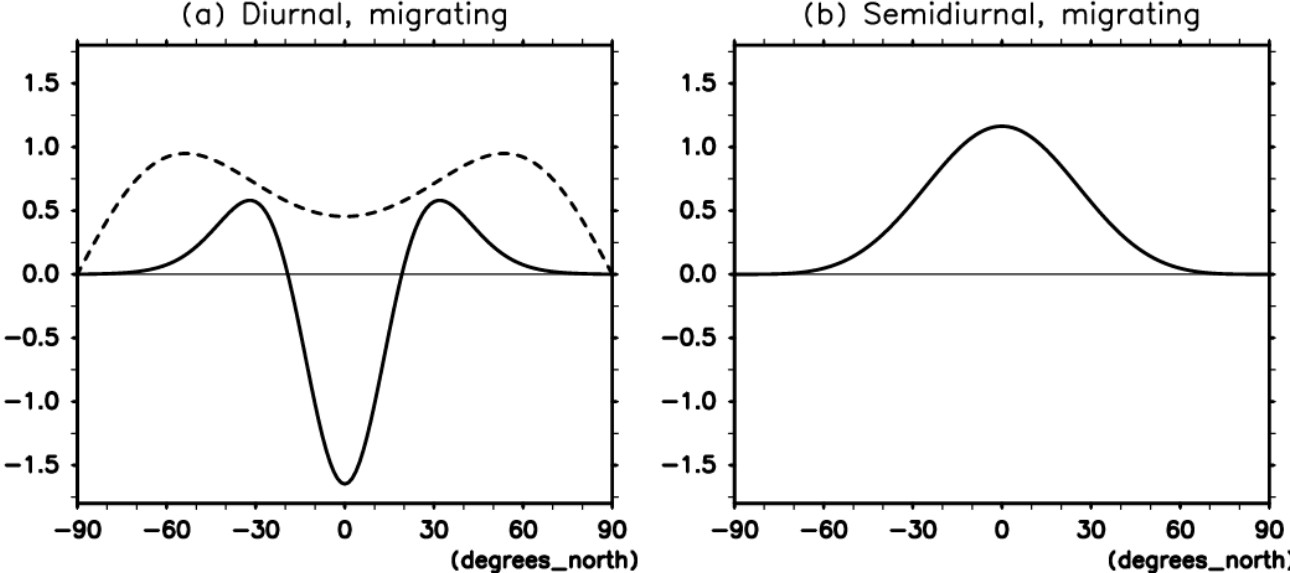

**Figure A1**: Meridional structure of Hough modes for (a) diurnal ($S_1$) migrating temperature tides (westward-propagating, zonal wavenumber 1, diurnal component) and (b) semidiurnal ($S_2$) migrating temperature tides (westward-propagating, zonal wavenumber 2, semidiurnal component). For (a), the leading gravest (solid) and trapped (dashed) modes are shown, while for (b) the gravest symmetric mode is shown.