# Peer review of "Representation of solar tides in the stratosphere and lower mesosphere in state-of-the-art reanalyses and in satellite observations"

_Atmospheric Chemistry and Physics, 2017_

## Referee Comment (RC1) · Anonymous Referee #1 · 4 Oct 2017

The authors performed a minor revision during the quick review process, and the article is in a very good shape now. The analysis of the solar tides in various data sets is very accurate and detailed. Thus, the article is a progress in this research area and of high interest for the readers of ACP. I only found a shortcoming in the figure caption 1 where the viewgraph g) is not mentioned.

---

## Referee Comment (RC2) · Anonymous Referee #3 · 5 Oct 2017

The paper presents a comparison of tidal components in the USLM in five different reanalyses datasets. Overall the paper is very well written, adequately illustrated, the analysis methods are valid, the literature is properly cited, and the conclusions are sound. I recommend publication with only the following minor revisions.

Minor comments:

Abstract: 1) Are SABER tidal amplitudes larger than reanalyses (and MLS) due to increased vertical resolution?

2) Can guidance be given advising which (if any) reanalyses should be used for tidal studies (especially those spanning multiple decades)?

2.2.1 SABER: Why not use the latest version 2 data?

2.2.2 Aura/MLS: Why not use the latest version 4 data?

4.3 Nonmigrating tides (Figure 12 and 13) 1) "All data sets show clear gravity-wave patterns being excited and emanating from the two major continents, namely, Africa (10-40°E) and South America (80-40°W) and also indicate a somewhat weaker wave source from the Maritime Continent (90-150°E)" – This is not clear to me from this figure. Clarify or edit accordingly.

2) I am also confused why the authors are saying that longitudinal variations in the nonmigrating tides are gravity waves. I am skeptical that the term "gravity waves" is correct. In the discussion of Figures 12 and 13, please either remove reference to "gravity waves" or defend in more detail.

3) "Westward (eastward) tilting waves correspond to the westward (eastward) propagating waves which are clear in the western (eastern) hemisphere." – Why is this? Instead of the waves propagating in different directions in the Eastern vs. Western hemispheres, could it be that there is a weak temperature ridge (positive region) extending upward from ~120E? Maybe this reflects trapped waves near the Asian monsoon? Either speculate a mechanism for different propagation directions or offer an alternative explanation.

5. Interannual Variations and Long-term Trends in Reanalysis representation of Tides Page 13, Paragraph 30: To be precise, Fujiwara et al. [2017] Figure 8 shows that the five reanalyses began assimilating AMSU-A in 1999. AMSU-B was assimilated in 1999 (MERRA and MERRA2), 2000 (CFSR), and 2001 (JRA-55 and ERA-I).
* * *

---

## Referee Comment (RC3) · Anonymous Referee #2 · 16 Oct 2017

**Review of** "Representation of solar tides in the stratosphere and lower mesosphere in state-of-the-art reanalyses and in satellite observations" (revised)

by T. Sakazaki et al.

**Recommendation:** Accept after minor revision

The authors responded satisfactorily to most of my initial comments. The more detailed explanation of how the migrating and non-migrating tides are extracted from satellite observations and reanalysis output is particularly welcome.

However, there is one question, about the interpretation of the phase behavior displayed in Figure 12, that was not addressed in the revised paper (see below). It would be helpful to have some clarification of this issue.

**Specific Comments (page, line):**

(11, 28) "Westward (eastward) tilting waves": The statement in the text appears to imply that, somehow, westward tilting waves dominate the wave field in the western hemisphere, while eastward-tilting waves are dominant in the eastern hemisphere. As noted in my original review, this is puzzling and calls for some explanation because, if these waves are being forced by the daily cycle of convection over land, both westward and eastward waves should be excited at each center of convection. Perhaps the appearance of predominant westward (eastward) tilt in the western (eastern) hemisphere is simply an artifact of the superposition of wave trains emanating from each center of convection? In any case, some explanatory remark about how this pattern might arise would be welcome here.

---

## Author Comment (AC1) · 5 Dec 2017

Authors' response to the Reviewer #1 comments on "Representation of solar tides in the stratosphere and lower mesosphere in state-of-the-art reanalyses and in satellite observations" by T. Sakazaki et al.

We appreciate the efforts of the Reviewer to evaluate our manuscript. Below we provide our specific responses to each of the Reviewer's points. Below we reproduce the Reviewer's comments verbatim in a **blue font** while our responses are in a standard **black font**.

The authors performed a minor revision during the quick review process, and the article is in a very good shape now. The analysis of the solar tides in various data sets is very accurate and detailed. Thus, the article is a progress in this research area and of high interest for the readers of ACP. I only found a shortcoming in the figure caption 1 where the viewgraph g) is not mentioned.

Thank you very much for favorable comments. The caption of Fig. 1 will be revised as:

*"Latitude-altitude distribution of amplitude for diurnal ($S_1$) migrating tide in temperature, as derived from (a) SABER, (b) JRA55, (c) JRA55-C, (d) JRA55-AMIP, (e) MERRA-2, (f) MERRA, (g) ERA-Interim, and (h) CFSR."*

---

## Author Comment (AC2) · 5 Dec 2017

Authors' response to the Reviewer #2 comments on "Representation of solar tides in the stratosphere and lower mesosphere in state-of-the-art reanalyses and in satellite observations" by T. Sakazaki et al.

We appreciate the efforts of the Reviewer to evaluate our manuscript. Below we provide our specific responses to each of the Reviewer's points. Below we reproduce the Reviewer's comments verbatim in a **blue font** while our responses are in a standard **black font**. We point to changes in the manuscript with line numbers in **red font** referring to the version of the manuscript with tracked changes shown.

**Recommendation:** Accept after minor revision
The authors responded satisfactorily to most of my initial comments. The more detailed explanation of how the migrating and non-migrating tides are extracted from satellite observations and reanalysis output is particularly welcome. However, there is one question, about the interpretation of the phase behavior displayed in Figure 12 that was not addressed in the revised paper (see below). It would be helpful to have some clarification of this issue.

Thank you very much for the favorable comments. Please see below for our answers to your specific comments.

**Specific Comments (page, line):**
(11, 28) "Westward (eastward) tilting waves": The statement in the text appears to imply that, somehow, westward tilting waves dominate the wave field in the western hemisphere, while eastward-tilting waves are dominant in the eastern hemisphere. As noted in my original review, this is puzzling and calls for some explanation because, if these waves are being forced by the daily cycle of convection over land, both westward and eastward waves should be excited at each center of convection. Perhaps the appearance of predominant westward (eastward) tilt in the western (eastern) hemisphere is simply an artifact of the superposition of wave trains emanating from each center of convection? In any case, some explanatory remark about how this pattern might arise would be welcome here.

We think that such asymmetry may be explained by two factors:
1) The major excitation regions are confined around -60 to +20°E (see also Sakazaki et al., 2015b); because waves are subject to dissipation during the horizontal propagation, westward waves are likely dominant to the west of -60°E and eastward waves are dominant to the east of 20°E.
2) Westward signals are clearer between -60 to +20°E, even though in this region both westward waves (from Africa) and eastward waves (from South-America) might be equally important. This asymmetry is likely because westward waves (mainly wavenumber 5) is more efficiently excited by

tropospheric heating than eastward waves (mainly wavenumber 3) (see also Fig. 15), due to the difference in their typical vertical wavelengths.

We have added these discussions in the revised manuscript (Page 11 L31- Page 12 L8).

---

## Author Comment (AC3) · 5 Dec 2017

Authors' response to the Reviewer #3 comments on "Representation of solar tides in the stratosphere and lower mesosphere in state-of-the-art reanalyses and in satellite observations" by T. Sakazaki et al.

We appreciate the efforts of the Reviewer to evaluate our manuscript. Below we provide our specific responses to each of the Reviewer's points. Below we reproduce the Reviewer's comments verbatim in a **blue font** while our responses are in a standard **black font**. We point to changes in the manuscript with line numbers in **red font** referring to the version of the manuscript with tracked changes shown.

The paper presents a comparison of tidal components in the USLM in five different reanalyses datasets. Overall the paper is very well written, adequately illustrated, the analysis methods are valid, the literature is properly cited, and the conclusions are sound. I recommend publication with only the following minor revisions. Minor comments:

Thank you very much for the favorable comments. Please see below for detailed responses.

Abstract:
1) Are SABER tidal amplitudes larger than reanalyses (and MLS) due to increased vertical resolution?

This might be partially responsible; but considering that only the diurnal migrating tide shows such a marked difference, we suspect that the vertical resolution may not be a major reason.

2) Can guidance be given advising which (if any) reanalyses should be used for tidal studies (especially those spanning multiple decades)?

As found in this study, the representation of tides is basically similar between the data sets at least during 2006-2012. In contrast, for the study of decadal variation in tides, our message may be (unfortunately) "do not use reanalyses for trend studies on tides" because all reanalyses show, more or less, artificial discontinuities due to the evolution of satellite observing systems as shown in Figs. 16-18 (Section 5). Other recent studies also reported such discontinuities in the stratosphere (Kawatani et al., 2016 for QBO; Long et al., 2017 for climatology).

2.2.1 SABER: Why not use the latest version 2 data?

As noted in (Page 5, L16), we used version 2.0 SABER data throughout the manuscript.

Thank you for the suggestion. We have updated Figure 5 by including version 4.2 MLS data; please see the figure below. It is confirmed that there is no major difference in the results between version 3.3 and version 4.2. In the revised manuscript, we will replace Fig. 5 by this Figure (Fig. R1).

[Figure]

**Figure R1 (revised Fig.5):** Vertical profile of difference between ascending and descending profiles of temperature in MLS measurements compared to the 1345 minus 0145 LT temperature difference sampled from the SABER data and various reanalysis data sets. For MLS, solid and dashed curves show the results from v4.2 and v3.3, respectively. Horizontal bars (only for MLS (v4) and SABER) show 95% confidential levels with t test. Results are annual means for the 7 year period 2006-2012.

4.3 Nonmigrating tides (Figure 12 and 13)
1) "All data sets show clear gravity-wave patterns being excited and emanating from the two major continents, namely, Africa (10-40°E) and South America (80-40°W) and also indicate a somewhat weaker wave source from the Maritime Continent (90-150°E)" – This is not clear to me from this figure. Clarify or edit accordingly.

It is seen that through the upper troposphere to the lower stratosphere, the wave signals are the strongest around South America and Africa, and both westward and eastward tilting waves emanate from there. This clearly demonstrates that these two continents are the main wave sources. Sakazaki et al. (2015b) also provided detailed discussions about this feature. In the revised manuscript, we have added the following description (Page 11 L24- L29):

*"It is clear that through the upper troposphere to the lower stratosphere, the wave signals are the strongest around the South America (80-40°W) and Africa (10-40°E) and are the second largest around the Maritime continent (90-150°E), and that both westward and eastward tilting waves emanate from these locations. This horizontal pattern indicates that nonmigrating tides are interpreted as the superposition of gravity waves from these geographically localized sources,*

*consistent with the finding by Sakazaki et al. (2015b, their section 4), who analyzed data from a high-resolution GCM as well as SABER and COSMIC GPS radio occultation measurements."*

2) I am also confused why the authors are saying that longitudinal variations in the nonmigrating tides are gravity waves. I am skeptical that the term "gravity waves" is correct. In the discussion of Figures 12 and 13, please either remove reference to "gravity waves" or defend in more detail.

We consider that tides are actually one kind of gravity waves. In the revised manuscript, we use the term, gravity-wave, in a more specific way like, *"the gravity-wave pattern emanating from the continents"* (e.g., Page 12, L25).

3) "Westward (eastward) tilting waves correspond to the westward (eastward) propagating waves which are clear in the western (eastern) hemisphere." – Why is this? Instead of the waves propagating in different directions in the Eastern vs. Western hemispheres, could it be that there is a weak temperature ridge (positive region) extending upward from 120E? Maybe this reflects trapped waves near the Asian monsoon? Either speculate a mechanism for different propagation directions or offer an alternative explanation.

We think that such asymmetry may be explained by two factors:
1) The major excitation regions are confined around -60 to +20°E (see also Sakazaki et al., 2015b); because waves are subject to dissipation during the horizontal propagation, westward waves are likely dominant to the west of -60°E and eastward waves are dominant to the east of 20°E.
2) Westward signals are clearer between -60 to +20°E, even though in this region both westward waves (from Africa) and eastward waves (from South-America) might be equally important. This asymmetry is likely because westward waves (mainly wavenumber 5) is more efficiently excited by tropospheric heating than eastward waves (mainly wavenumber 3) (see also Fig. 15), due to the difference in their typical vertical wavelengths.

We have added these discussions in the revised manuscript (Page 11 L31- Page 12 L8).

5. Interannual Variations and Long-term Trends in Reanalysis representation of Tides Page 13, Paragraph 30: To be precise, Fujiwara et al. [2017] Figure 8 shows that the five reanalyses began assimilating AMSU-A in 1999. AMSU-B was assimilated in 1999 (MERRA and MERRA2), 2000 (CFSR), and 2001 (JRA-55 and ERA-I).

Thank you for the clarification. We mention this simply by revising the manuscript (Page 14, L10-13) as,

*"ATOVS has the AMSU-A/B, which has more channels in the upper stratosphere…(see Fujiwara et al., 2017, their section 5.2 for more details)."*

In this paper, we would like to avoid detailed discussion about whether AMSU-A or –B is related.